# Parallelizing MCMC with Random Partition Trees

**Xiangyu Wang**
Dept. of Statistical Science
Duke University
xw56@stat.duke.edu

**Fangjian Guo**
Dept. of Computer Science
Duke University
guo@cs.duke.edu

**Katherine A. Heller**
Dept. of Statistical Science
Duke University
kheller@stat.duke.edu

**David B. Dunson**
Dept. of Statistical Science
Duke University
dunson@stat.duke.edu

## Abstract

The modern scale of data has brought new challenges to Bayesian inference. In particular, conventional MCMC algorithms are computationally very expensive for large data sets. A promising approach to solve this problem is embarrassingly parallel MCMC (EP-MCMC), which first partitions the data into multiple subsets and runs independent sampling algorithms on each subset. The subset posterior draws are then aggregated via some combining rules to obtain the final approximation. Existing EP-MCMC algorithms are limited by approximation accuracy and difficulty in resampling. In this article, we propose a new EP-MCMC algorithm *PART* that solves these problems. The new algorithm applies *random partition trees* to combine the subset posterior draws, which is distribution-free, easy to resample from and can adapt to multiple scales. We provide theoretical justification and extensive experiments illustrating empirical performance.

## 1 Introduction

Bayesian methods are popular for their success in analyzing complex data sets. However, for large data sets, Markov Chain Monte Carlo (MCMC) algorithms, widely used in Bayesian inference, can suffer from huge computational expense. With large data, there is increasing time per iteration, increasing time to convergence, and difficulties with processing the full data on a single machine due to memory limits. To ameliorate these concerns, various methods such as stochastic gradient Monte Carlo [1] and sub-sampling based Monte Carlo [2] have been proposed. Among directions that have been explored, embarrassingly parallel MCMC (EP-MCMC) seems most promising. EP-MCMC algorithms typically divide the data into multiple subsets and run independent MCMC chains simultaneously on each subset. The posterior draws are then aggregated according to some rules to produce the final approximation. This approach is clearly more efficient as now each chain involves a much smaller data set and the sampling is communication-free. The key to a successful EP-MCMC algorithm lies in the speed and accuracy of the combining rule.

Existing EP-MCMC algorithms can be roughly divided into three categories. The first relies on asymptotic normality of posterior distributions. [3] propose a "Consensus Monte Carlo" algorithm, which produces final approximation by a weighted averaging over all subset draws. This approach is effective when the posterior distributions are close to Gaussian, but could suffer from huge bias when skewness and multi-modes are present. The second category relies on calculating an appropriate variant of a mean or median of the subset posterior measures [4, 5]. These approaches rely on asymptotics (size of data increasing to infinity) to justify accuracy, and lack guarantees in finite samples. The third category relies on the *product density equation (PDE)* in (1). Assuming $X$ is the

observed data and $\theta$ is the parameter of interest, when the observations are iid conditioned on $\theta$, for any partition of $X = X^{(1)} \cup X^{(2)} \cup \cdots \cup X^{(m)}$, the following identity holds,

$$p(\theta|X) \propto \pi(\theta)p(X|\theta) \propto p(\theta|X^{(1)})p(\theta|X^{(2)}) \cdots p(\theta|X^{(m)}), \tag{1}$$

if the prior on the full data and subsets satisfy $\pi(\theta) = \prod_{i=1}^{m} \pi_i(\theta)$. [6] proposes using kernel density estimation on each subset posterior and then combining via (1). They use an independent Metropolis sampler to resample from the combined density. [7] apply the Weierstrass transform directly to (1) and developed two sampling algorithms based on the transformed density. These methods guarantee the approximation density converges to the true posterior density as the number of posterior draws increase. However, as both are kernel-based, the two methods are limited by two major drawbacks. The first is the inefficiency of resampling. Kernel density estimators are essentially mixture distributions. Assuming we have collected 10,000 posterior samples on each machine, then multiplying just two densities already yields a mixture distribution containing $10^8$ components, each of which is associated with a different weight. The resampling requires the independent Metropolis sampler to search over an exponential number of mixture components and it is likely to get stuck at one "good" component, resulting in high rejection rates and slow mixing. The second is the sensitivity to bandwidth choice, with one bandwidth applied to the whole space.

In this article, we propose a novel EP-MCMC algorithm termed "parallel aggregation random trees" (*PART*), which solves the above two problems. The algorithm inhibits the explosion of mixture components so that the aggregated density is easy to resample. In addition, the density estimator is able to adapt to multiple scales and thus achieve better approximation accuracy. In Section 2, we motivate the new methodology and present the algorithm. In Section 3, we present error bounds and prove consistency of *PART* in the number of posterior draws. Experimental results are presented in Section 4. Proofs and part of the numerical results are provided in the supplementary materials.

## 2 Method

Recall the *PDE* identity (1) in the introduction. When data set $X$ is partitioned into $m$ subsets $X = X^{(1)} \cup \cdots \cup X^{(m)}$, the posterior distribution of the $i^{\text{th}}$ subset can be written as

$$f^{(i)}(\theta) \propto \pi(\theta)^{1/m}p(X^{(i)}|\theta), \tag{2}$$

where $\pi(\theta)$ is the prior assigned to the full data set. Assuming observations are iid given $\theta$, the relationship between the full data posterior and subset posteriors is captured by

$$p(\theta|X) \propto \pi(\theta) \prod_{i=1}^{m} p(X^{(i)}|\theta) \propto \prod_{i=1}^{m} f^{(i)}(\theta). \tag{3}$$

Due to the flaws of applying kernel-based density estimation to (3) mentioned above, we propose to use *random partition trees* or *multi-scale histograms*. Let $\mathcal{F}_K$ be the collection of all $\mathbb{R}^p$-partitions formed by $K$ disjoint rectangular blocks, where a rectangular block takes the form of $A_k \overset{def}{=} (l_{k,1}, r_{k,1}] \times (l_{k,2}, r_{k,2}] \times \cdots (l_{k,p}, r_{k,p}] \subseteq \mathbb{R}^p$ for some $l_{k,q} < r_{k,q}$. A $K$-block histogram is then defined as

$$\hat{f}^{(i)}(\theta) = \sum_{k=1}^{K} \frac{n_k^{(i)}}{N|A_k|}\mathbf{1}(\theta \in A_k), \tag{4}$$

where $\{A_k : k = 1, 2, \cdots, K\} \in \mathcal{F}_K$ are the blocks and $N, n_k^{(i)}$ are the total number of posterior samples on the $i^{\text{th}}$ subset and of those inside the block $A_k$ respectively (assuming the same $N$ across subsets). We use $|\cdot|$ to denote the area of a block. Assuming each subset posterior is approximated by a $K$-block histogram, if the partition $\{A_k\}$ is restricted to be *the same* across all subsets, then the aggregated density after applying (3) is still a $K$-block histogram (illustrated in the supplement),

$$\hat{p}(\theta|X) = \frac{1}{Z} \prod_{i=1}^{m} \hat{f}^{(i)}(\theta) = \frac{1}{Z} \sum_{k=1}^{K} \left( \prod_{i=1}^{m} \frac{n_k^{(i)}}{|A_k|} \right) \mathbf{1}(\theta \in A_k) = \sum_{k=1}^{K} w_k g_k(\theta), \tag{5}$$

where $Z = \sum_{k=1}^{K} \prod_{i=1}^{m} n_k^{(i)}/|A_k|^{m-1}$ is the normalizing constant, $w_k$'s are the updated weights, and $g_k(\theta) = \text{unif}(\theta; A_k)$ is the block-wise distribution. Common histogram blocks across subsets control the number of mixture components, leading to simple aggregation and resampling procedures. Our *PART* algorithm consists of *space partitioning* followed by *density aggregation*, with aggregation simply multiplying densities across subsets for each block and then normalizing.

## 2.1 Space Partitioning

To find good partitions, our algorithm recursively bisects (not necessarily evenly) a previous block along a randomly selected dimension, subject to certain rules. Such partitioning is multi-scale and related to wavelets [8]. Assume we are currently splitting the block $A$ along the dimension $q$ and denote the posterior samples in $A$ by $\{\theta_j^{(i)}\}_{j \in A}$ for the $i^{\text{th}}$ subset. The cut point on dimension $q$ is determined by a partition rule $\phi(\{\theta_{j,q}^{(1)}\}, \{\theta_{j,q}^{(2)}\}, \cdots, \{\theta_{j,q}^{(m)}\})$. The resulting two blocks are subject to further bisecting under the same procedure until one of the following stopping criteria is met — (i) $n_k/N < \delta_\rho$ or (ii) the area of the block $|A_k|$ becomes smaller than $\delta_{|A|}$. The algorithm returns a tree with $K$ leafs, each corresponding to a block $A_k$. Details are provided in Algorithm 1.

---

**Algorithm 1** Partition tree algorithm

---

1: **procedure** BUILDTREE($\{\theta_j^{(1)}\}, \{\theta_j^{(2)}\}, \cdots, \{\theta_j^{(m)}\}, \phi(\cdot), \delta_\rho, \delta_a, N, L, R$)
2:   $D \leftarrow \{1, 2, \cdots, p\}$
3:   **while** $D$ not empty **do**
4:     Draw $q$ uniformly at random from $D$.          ▷ Randomly choose the dimension to cut
5:     $\theta_q^* \leftarrow \phi(\{\theta_{j,q}^{(1)}\}, \{\theta_{j,q}^{(2)}\}, \cdots, \{\theta_{j,q}^{(m)}\})$,   $\mathcal{T}.n^{(i)} \leftarrow$ Cardinality of $\{\theta_j^{(i)}\}$ for all $i$
6:     **if** $\theta_q^* - L_q > \delta_a$, $R_q - \theta_q^* > \delta_a$ **and** $\min(\sum_j \mathbf{1}(\theta_{j,q}^{(i)} \leq \theta_q^*), \sum_j \mathbf{1}(\theta_{j,q}^{(i)} > \theta_q^*)) > N\delta_\rho$
    for all $i$ **then**
7:       $L' \leftarrow L, L'_q \leftarrow \theta_q^*, R' \leftarrow R, R'_q \leftarrow \theta_q^*$          ▷ Update left and right boundaries
8:       $\mathcal{T}.\mathcal{L} \leftarrow$ BUILDTREE($\{\theta_j^{(1)} : \theta_{j,q}^{(1)} \leq \theta_q^*\}, \cdots, \{\theta_j^{(m)} : \theta_{j,q}^{(m)} \leq \theta_q^*\}, \cdots, N, L, R'$)
9:       $\mathcal{T}.\mathcal{R} \leftarrow$ BUILDTREE($\{\theta_j^{(1)} : \theta_{j,q}^{(1)} > \theta_q^*\}, \cdots, \{\theta_j^{(m)} : \theta_{j,q}^{(m)} > \theta_q^*\}, \cdots, N, L', R$)
10:      **return** $\mathcal{T}$
11:    **else**
12:      $D \leftarrow D \setminus \{q\}$          ▷ Try cutting at another dimension
13:    **end if**
14:  **end while**
15:  $\mathcal{T}.\mathcal{L} \leftarrow$ NULL, $\mathcal{T}.\mathcal{R} \leftarrow$ NULL, **return** $\mathcal{T}$          ▷ Leaf node
16: **end procedure**

---

In Algorithm 1, $\delta_{|A|}$ becomes the minimum edge length of a block $\delta_a$ (possibly different across dimensions). Quantities $L, R \in \mathbb{R}^p$ are the left and right boundaries of the samples respectively, which take the sample minimum/maximum when the support is unbounded. We consider two choices for the partition rule $\phi(\cdot)$ — maximum (empirical) likelihood partition (ML) and median/KD-tree partition (KD).

**Maximum Likelihood Partition (ML)**   ML-partition searches for partitions by greedily maximizing the empirical log likelihood at each iteration. For $m = 1$ we have

$$\theta^* = \phi_{\text{ML}}(\{\theta_{j,q}, j = 1, \cdots, n\}) = \underset{n_1 + n_2 = n, A_1 \cup A_2 = A}{\arg\max} \left(\frac{n_1}{n|A_1|}\right)^{n_1} \left(\frac{n_2}{n|A_2|}\right)^{n_2}, \quad (6)$$

where $n_1$ and $n_2$ are counts of posterior samples in $A_1$ and $A_2$, respectively. The solution to (6) falls inside the set $\{\theta_j\}$. Thus, a simple linear search after sorting samples suffices (by book-keeping the ordering, sorting the whole block once is enough for the entire procedure). For $m > 1$, we have

$$\phi_{q,\text{ML}}(\cdot) = \underset{\theta^* \in \cup_{i=1}^m \{\theta_j^{(i)}\}}{\arg\max} \prod_{i=1}^m \left(\frac{n_1^{(i)}}{n^{(i)}|A_1|}\right)^{n_1^{(i)}} \left(\frac{n_2^{(i)}}{n^{(i)}|A_2|}\right)^{n_2^{(i)}}, \quad (7)$$

similarly solved by a linear search. This is dominated by sorting and takes $O(n \log n)$ time.

**Median/KD-Tree Partition (KD)**   Median/KD-tree partition cuts at the empirical median of posterior samples. When there are multiple subsets, the median is taken over pooled samples to force $\{A_k\}$ to be the same across subsets. Searching for median takes $O(n)$ time [9], which is faster than ML-partition especially when the number of posterior draws is large. The same partitioning strategy is adopted by KD-trees [10].

## 2.2 Density Aggregation

Given a common partition, Algorithm 2 aggregates all subsets *in one stage*. However, assuming a single "good" partition for all subsets is overly restrictive when $m$ is large. Hence, we also consider *pairwise aggregation* [6, 7], which recursively groups subsets into pairs, combines each pair with Algorithm 2, and repeats until one final set is obtained. Run time of *PART* is dominated by space partitioning (BUILDTREE), with normalization and resampling very fast.

---

**Algorithm 2** Density aggregation algorithm (drawing $N'$ samples from the aggregated posterior)

---

1: **procedure** ONESTAGEAGGREGATE($\{\theta_j^{(1)}\}, \{\theta_j^{(2)}\}, \cdots, \{\theta_j^{(m)}\}, \phi(\cdot), \delta_\rho, \delta_a, N, N', L, R$)
2:     $\mathcal{T} \leftarrow$ BUILDTREE($\{\theta_j^{(1)}\}, \{\theta_j^{(2)}\}, \cdots, \{\theta_j^{(m)}\}, \phi(\cdot), \delta_\rho, \delta_a, N, L, R$),    $Z \leftarrow 0$
3:     $(\{A_k\}, \{n_k^{(i)}\}) \leftarrow$ TRAVERSELEAF($\mathcal{T}$)
4:     **for** $k = 1, 2, \cdots, K$ **do**
5:         $\tilde{w}_k \leftarrow \prod_{i=1}^m n_k^{(i)} / |A_k|^{m-1}, Z \leftarrow Z + \tilde{w}_k$             ▷ Multiply inside each block
6:     **end for**
7:     $w_k \leftarrow \tilde{w}_k / Z$ for all $k$                                             ▷ Normalize
8:     **for** $t = 1, 2, \cdots, N'$ **do**
9:         Draw $k$ with weights $\{w_k\}$ and then draw $\theta_t \sim g_k(\theta)$
10:     **end for**
11:     **return** $\{\theta_1, \theta_2, \cdots, \theta_{N'}\}$
12: **end procedure**

---

## 2.3 Variance Reduction and Smoothing

**Random Tree Ensemble**   Inspired by random forests [11, 12], the full posterior is estimated by averaging $T$ independent trees output by Algorithm 1. Smoothing and averaging can reduce variance and yield better approximation accuracy. The trees can be built in parallel and resampling in Algorithm 2 only additionally requires picking a tree uniformly at random.

**Local Gaussian Smoothing**   As another approach to increase smoothness, the blockwise uniform distribution in (5) can be replaced by a Gaussian distribution $g_k = N(\theta; \mu_k, \Sigma_k)$, with mean and covariance estimated "locally" by samples within the block. A multiplied Gaussian approximation is used: $\Sigma_k = (\sum_{i=1}^m \hat{\Sigma}_k^{(i)-1})^{-1}, \mu_k = \Sigma_k(\sum_{i=1}^m \hat{\Sigma}_k^{(i)-1} \hat{\mu}_k^{(i)})$, where $\hat{\Sigma}_k^{(i)}$ and $\hat{\mu}_k^{(i)}$ are estimated with the $i^{\text{th}}$ subset. We apply both random tree ensembles and local Gaussian smoothing in all applications of PART in this article unless explicitly stated otherwise.

# 3 Theory

In this section, we provide consistency theory (in the number of posterior samples) for histograms and the aggregated density. We do not consider the variance reduction and smoothing modifications in these developments for simplicity in exposition, but extensions are possible. Section 3.1 provides error bounds on ML and KD-tree partitioning-based histogram density estimators constructed from $N$ independent samples from a single joint posterior; modified bounds can be obtained for MCMC samples incorporating the mixing rate, but will not be considered here. Section 3.2 then provides corresponding error bounds for our PART-aggregated density estimators in the one-stage and pairwise cases. Detailed proofs are provided in the supplementary materials.

Let $f(\theta)$ be a $p$-dimensional posterior density function. Assume $f$ is supported on a measurable set $\Omega \subseteq \mathbb{R}^p$. Since one can always transform $\Omega$ to a bounded region by scaling, we simply assume $\Omega = [0,1]^p$ as in [8, 13] without loss of generality. We also assume that $f \in C^1(\Omega)$.

## 3.1 Space partitioning

**Maximum likelihood partition (ML)**   For a given $K$, ML partition solves the following problem:

$$\hat{f}_{ML} = \arg\max \frac{1}{N} \sum_{k=1}^K n_k \log\left(\frac{n_k}{N|A_k|}\right), \quad \text{s.t. } n_k/N \geq c_0\rho, \ |A_k| \geq \rho/D, \qquad (8)$$

for some $c_0$ and $\rho$, where $D = \|f\|_\infty < \infty$. We have the following result.

**Theorem 1.** *Choose $\rho = 1/K^{1+1/(2p)}$. For any $\delta > 0$, if the sample size satisfies that $N > 2(1-c_0)^{-2}K^{1+1/(2p)}\log(2K/\delta)$, then with probability at least $1 - \delta$, the optimal solution to (8) satisfies that*

$$D_{KL}(f\|\hat{f}_{ML}) \leq (C_1 + 2\log K)K^{-\frac{1}{2p}} + C_2 \max\left\{\log D, 2\log K\right\}\sqrt{\frac{K}{N}\log\left(\frac{3eN}{K}\right)\log\left(\frac{8}{\delta}\right)},$$

*where $C_1 = \log D + 4pLD$ with $L = \|f'\|_\infty$ and $C_2 = 48\sqrt{p+1}$.*

When multiple densities $f^{(1)}(\theta), \cdots, f^{(m)}(\theta)$ are presented, our goal of imposing the same partition on all functions requires solving a different problem,

$$(\hat{f}_{ML}^{(i)})_{i=1}^m = \arg\max \sum_{i=1}^m \frac{1}{N_i}\sum_{k=1}^K n_k^{(i)}\log\left(\frac{n_k^{(i)}}{N_i|A_k|}\right), \quad \text{s.t. } n_k^{(i)}/N_i \geq c_0\rho, \; |A_k| \geq \rho/D, \quad (9)$$

where $N_i$ is the number of posterior samples for function $f^{(i)}$. A similar result as Theorem 1 for (9) is provided in the supplementary materials.

**Median partition/KD-tree (KD)** The KD-tree $\hat{f}_{KD}$ cuts at the empirical median for different dimensions. We have the following result.

**Theorem 2.** *For any $\varepsilon > 0$, define $r_\varepsilon = \log_2\left(1 + \frac{1}{2+3L/\varepsilon}\right)$. For any $\delta > 0$, if $N > 32e^2(\log K)^2 K\log(2K/\delta)$, then with probability at least $1 - \delta$, we have*

$$\|\hat{f}_{KD} - f_{KD}\|_1 \leq \varepsilon + pLK^{-r_\varepsilon/p} + 4e\log K\sqrt{\frac{2K}{N}\log\left(\frac{2K}{\delta}\right)}.$$

*If $f(\theta)$ is further lower bounded by some constant $b_0 > 0$, we can then obtain an upper bound on the KL-divergence. Define $r_{b_0} = \log_2\left(1 + \frac{1}{2+3L/b_0}\right)$ and we have*

$$D_{KL}(f\|\hat{f}_{KD}) \leq \frac{pLD}{b_0}K^{-r_{b_0}/p} + 8e\log K\sqrt{\frac{2K}{N}\log\left(\frac{2K}{\delta}\right)}.$$

When there are multiple functions and the median partition is performed on pooled data, the partition might not happen at the empirical median on each subset. However, as long as the partition quantiles are upper and lower bounded by $\alpha$ and $1 - \alpha$ for some $\alpha \in [1/2, 1)$, we can establish results similar to Theorem 2. The result is provided in the supplementary materials.

### 3.2 Posterior aggregation

The previous section provides estimation error bounds on individual posterior densities, through which we can bound the distance between the true posterior conditional on the full data set and the aggregated density via (3). Assume we have $m$ density functions $\{f^{(i)}, i = 1, 2, \cdots, m\}$ and intend to approximate their aggregated density $f_I = \prod_{i \in I} f^{(i)} / \int \prod_{i \in I} f^{(i)}$, where $I = \{1, 2, \cdots, m\}$. Notice that for any $I' \subseteq I$, $f_{I'} = p(\theta|\bigcup_{i \in I'} X^{(i)})$. Let $D = \max_{I' \subseteq I}\|f_{I'}\|_\infty$, i.e., $D$ is an upper bound on all posterior densities formed by a subset of $X$. Also define $Z_{I'} = \int \prod_{i \in I'} f^{(i)}$. These quantities depend only on the model and the observed data (not posterior samples). We denote $\hat{f}_{ML}$ and $\hat{f}_{KD}$ by $\hat{f}$ as the following results apply similarly to both methods.

The "one-stage" aggregation (Algorithm 2) first obtains an approximation for each $f^{(i)}$ (via either ML-partition or KD-partition) and then computes $\hat{f}_I = \prod_{i \in I} \hat{f}^{(i)} / \int \prod_{i \in I} \hat{f}^{(i)}$.

**Theorem 3** (One-stage aggregation)**.** *Denote the average total variation distance between $f^{(i)}$ and $\hat{f}^{(i)}$ by $\varepsilon$. Assume the conditions in Theorem 1 and 2 and for ML-partition*

$$\sqrt{N} \geq 32c_0^{-1}\sqrt{2(p+1)}K^{\frac{3}{2}+\frac{1}{2p}}\sqrt{\log\left(\frac{3eN}{K}\right)\log\left(\frac{8}{\delta}\right)}$$

*and for KD-partition*

$$N > 128e^2K(\log K)^2\log(K/\delta).$$

*Then with high probability the total variation distance between $f_I$ and $\hat{f}_I$ is bounded by $\|f_I - \hat{f}_I\|_1 \leq \frac{2}{Z_I}m(2D)^{m-1}\varepsilon$, where $Z_I$ is a constant that does not depend on the posterior samples.*

The approximation error of Algorithm 2 increases dramatically with the number of subsets. To ameliorate this, we introduce the *pairwise aggregation* strategy in Section 2, for which we have the following result.

**Theorem 4** (Pairwise aggregation)**.** *Denote the average total variation distance between $f^{(i)}$ and $\hat{f}^{(i)}$ by $\varepsilon$. Assume the conditions in Theorem 3. Then with high probability the total variation distance between $f_I$ and $\hat{f}_I$ is bounded by $\|f_I - \hat{f}_I\|_1 \leq (4C_0D)^{\log_2 m+1}\varepsilon$, where $C_0 = \max_{I'' \subset I' \subseteq I}\frac{Z_{I''}Z_{I'\setminus I''}}{Z_{I'}}$ is a constant that does not depend on posterior samples.*

## 4 Experiments

In this section, we evaluate the empirical performance of *PART*[1] and compare the two algorithms *PART-KD* and *PART-ML* to the following posterior aggregation algorithms.

1. **Simple averaging** (*average*): each aggregated sample is an arithmetic average of $M$ samples coming from $M$ subsets.

2. **Weighted averaging** (*weighted*): also called **Consensus Monte Carlo** algorithm [3], where each aggregated sample is a weighted average of $M$ samples. The weights are optimally chosen for a Gaussian posterior.

3. **Weierstrass rejection sampler** (*Weierstrass*): subset posterior samples are passed through a rejection sampler based on the Weierstrass transform to produce the aggregated samples [7]. We use its R package[2] for experiments.

4. **Parametric density product** (*parametric*): aggregated samples are drawn from a multivariate Gaussian, which is a product of Laplacian approximations to subset posteriors [6].

5. **Nonparametric density product** (*nonparametric*): aggregated posterior is approximated by a product of kernel density estimates of subset posteriors [6]. Samples are drawn with an independent Metropolis sampler.

6. **Semiparametric density product** (*semiparametric*): similar to the *nonparametric*, but with subset posteriors estimated semiparametrically [6, 14].

All experiments except the two toy examples use adaptive MCMC [15, 16] [3] for posterior sampling. For *PART-KD/ML*, one-stage aggregation (Algorithm 2) is used only for the toy examples (results from pairwise aggregation are provided in the supplement). For other experiments, pairwise aggregation is used, which draws 50,000 samples for intermediate stages and halves $\delta_\rho$ after each stage to refine the resolution (The value of $\delta_\rho$ listed below is for the final stage). The random ensemble of *PART* consists of 40 trees.

### 4.1 Two Toy Examples

The two toy examples highlight the performance of our methods in terms of (i) recovering multiple modes and (ii) correctly locating posterior mass when subset posteriors are heterogeneous. The *PART-KD/PART-ML* results are obtained from Algorithm 2 without local Gaussian smoothing.

**Bimodal Example**  Figure 1 shows an example consisting of $m = 10$ subsets. Each subset consists of 10,000 samples drawn from a mixture of two univariate normals $0.27N(\mu_{i,1}, \sigma_{i,1}^2) + 0.73N(\mu_{i,2}, \sigma_{i,2}^2)$, with the means and standard deviations slightly different across subsets, given by $\mu_{i,1} = -5 + \epsilon_{i,1}$, $\mu_{i,2} = 5 + \epsilon_{i,2}$ and $\sigma_{i,1} = 1 + |\delta_{i,1}|$, $\sigma_{i,2} = 4 + |\delta_{i,2}|$, where $\epsilon_{i,l} \sim N(0, 0.5)$, $\delta_{i,l} \sim N(0, 0.1)$ independently for $m = 1, \cdots, 10$ and $l = 1, 2$. The resulting true combined posterior (red solid) consists of two modes with different scales. In Figure 1, the left panel shows the subset posteriors (dashed) and the true posterior; the right panel compares the results with various methods to the truth. A few are omitted in the graph: *average* and *weighted average* overlap with *parametric*, and *Weierstrass* overlaps with *PART-KD/PART-ML*.

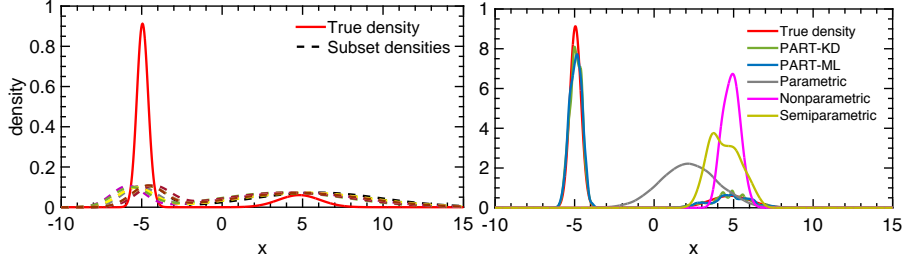

Figure 1: Bimodal posterior combined from 10 subsets. Left: the true posterior and subset posteriors (dashed). Right: aggregated posterior output by various methods compared to the truth. Results are based on 10,000 aggregated samples.

**Rare Bernoulli Example**  We consider $N = 10,000$ Bernoulli trials $x_i \overset{iid}{\sim} \text{Ber}(\theta)$ split into $m = 15$ subsets. The parameter $\theta$ is chosen to be $2m/N$ so that on average each subset only contains 2 successes. By random partitioning, the subset posteriors are rather heterogeneous as plotted in dashed lines in the left panel of Figure 2. The prior is set as $\pi(\theta) = \text{Beta}(\theta; 2, 2)$. The right panel of Figure 2 compares the results of various methods. *PART-KD*, *PART-ML* and *Weierstrass* capture the true posterior shape, while *parametric*, *average* and *weighted average* are all biased. The *nonparametric* and *semiparametric* methods produce flat densities near zero (not visible in Figure 2 due to the scale).

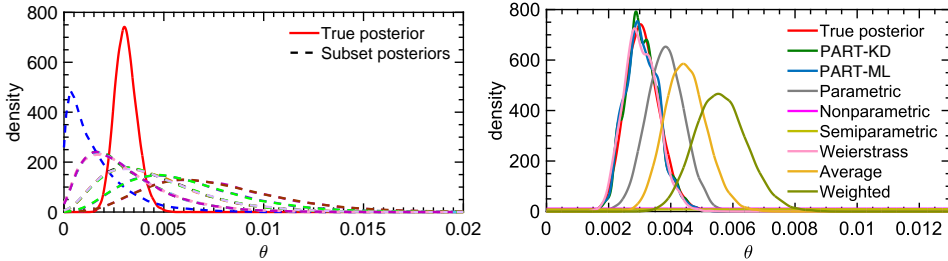

Figure 2: The posterior for the probability $\theta$ of a rare event. Left: the full posterior (solid) and $m = 15$ subset posteriors (dashed). Right: aggregated posterior output by various methods. All results are based on 20,000 aggregated samples.

## 4.2 Bayesian Logistic Regression

**Synthetic dataset**  The dataset $\{(\mathbf{x}_i, y_i)\}_{i=1}^N$ consists of $N = 50,000$ observations in $p = 50$ dimensions. All features $\mathbf{x}_i \in \mathbb{R}^{p-1}$ are drawn from $N_{p-1}(\mathbf{0}, \mathbf{\Sigma})$ with $p = 50$ and $\Sigma_{k,l} = 0.9^{|k-l|}$. The model intercept is set to $-3$ and the other coefficient $\theta_j^*$'s are drawn randomly from $N(0, 5^2)$. Conditional on $\mathbf{x}_i$, $y_i \in \{0, 1\}$ follows $p(y_i = 1) = 1/(1 + \exp(-\boldsymbol{\theta}^{*T}[1, \mathbf{x}_i]))$. The dataset is randomly split into $m = 40$ subsets. For both full chain and subset chains, we run adaptive MCMC for 200,000 iterations after 100,000 burn-in. Thinning by 4 results in $T = 50,000$ samples.

The samples from the full chain (denoted as $\{\boldsymbol{\theta}_j\}_{j=1}^T$) are treated as the ground truth. To compare the accuracy of different methods, we resample $T$ points $\{\hat{\boldsymbol{\theta}}_j\}$ from each aggregated posterior and then

compare them using the following metrics: (1) RMSE of posterior mean $\|\frac{1}{pT}(\sum_j \hat{\boldsymbol{\theta}}_j - \sum_j \boldsymbol{\theta}_j)\|_2$ (2) approximate KL divergence $D_{\text{KL}}(p(\boldsymbol{\theta})\|\hat{p}(\boldsymbol{\theta}))$ and $D_{\text{KL}}(\hat{p}(\boldsymbol{\theta})\|p(\boldsymbol{\theta}))$, where $\hat{p}$ and $p$ are both approximated by multivariate Gaussians (3) the posterior concentration ratio, defined as $r = \sqrt{\sum_j \|\hat{\boldsymbol{\theta}}_j - \boldsymbol{\theta}^*\|_2^2 / \sum_j \|\boldsymbol{\theta}_j - \boldsymbol{\theta}^*\|_2^2}$, which measures how posterior spreads out around the true value (with $r = 1$ being ideal). The result is provided in Table 1. Figure 4 shows the $D_{\text{KL}}(p\|\hat{p})$ versus the length of subset chains supplied to the aggregation algorithm. The results of *PART* are obtained with $\delta_\rho = 0.001$, $\delta_a = 0.0001$ and 40 trees. Figure 3 showcases the aggregated posterior for two parameters in terms of joint and marginal distributions.

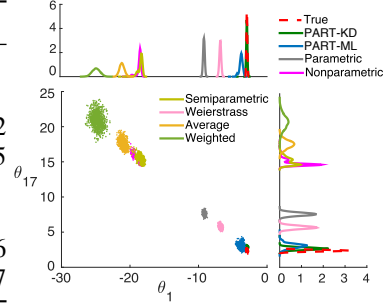

| Method | RMSE | $D_{\text{KL}}(p\|\hat{p})$ | $D_{\text{KL}}(\hat{p}\|p)$ | $r$ |
|---|---|---|---|---|
| PART (KD) | **0.587** | $3.95 \times 10^2$ | $6.45 \times 10^2$ | **3.94** |
| PART (ML) | 1.399 | $\mathbf{8.05 \times 10^1}$ | $\mathbf{5.47 \times 10^2}$ | 9.17 |
| average | 29.93 | $2.53 \times 10^3$ | $5.41 \times 10^4$ | 184.62 |
| weighted | 38.28 | $2.60 \times 10^4$ | $2.53 \times 10^5$ | 236.15 |
| Weierstrass | 6.47 | $7.20 \times 10^2$ | $2.62 \times 10^3$ | 39.96 |
| parametric | 10.07 | $2.46 \times 10^3$ | $6.12 \times 10^3$ | 62.13 |
| nonparametric | 25.59 | $3.40 \times 10^4$ | $3.95 \times 10^4$ | 157.86 |
| semiparametric | 25.45 | $2.06 \times 10^4$ | $3.90 \times 10^4$ | 156.97 |

Table 1: Accuracy of posterior aggregation on logistic regression. Figure 3: Posterior of $\theta_1$ and $\theta_{17}$.

**Real datasets**  We also run experiments on two real datasets: (1) the *Covertype* dataset[4] [17] consists of 581,012 observations in 54 dimensions, and the task is to predict the type of forest cover with cartographic measurements; (2) the *MiniBooNE* dataset[5] [18, 19] consists of 130,065 observations in 50 dimensions, whose task is to distinguish electron neutrinos from muon neutrinos with experimental data. For both datasets, we reserve $1/5$ of the data as the test set. The training set is randomly split into $m = 50$ and $m = 25$ subsets respectively for *covertype* and *MiniBooNE*. Figure 5 shows the prediction accuracy versus total runtime (parallel subset MCMC + aggregation time) for different methods. For each MCMC chain, the first 20% iterations are discarded before aggregation as burn-in. The aggregated chain is required to be of the same length as the subset chains. As a reference, we also plot the result for the full chain and *lasso* [20] run on the full training set.

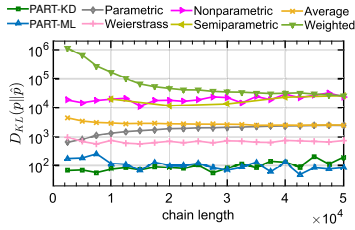

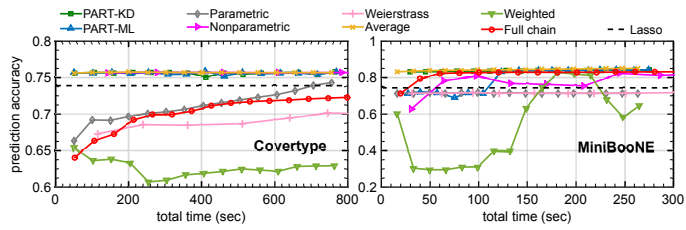

Figure 4: Approximate KL divergence between the full chain and the combined posterior versus the length of subset chains.

Figure 5: Prediction accuracy versus total runtime (running chain + aggregation) on *Covertype* and *MiniBooNE* datasets (*semiparametric* is not compared due to its long running time). Plots against the length of chain are provided in the supplement.

## 5   Conclusion

In this article, we propose a new embarrassingly-parallel MCMC algorithm *PART* that can efficiently draw posterior samples for large data sets. *PART* is simple to implement, efficient in subset combining and has theoretical guarantees. Compared to existing EP-MCMC algorithms, *PART* has substantially improved performance. Possible future directions include (1) exploring other multi-scale density estimators which share similar properties as partition trees but with a better approximation accuracy (2) developing a tuning procedure for choosing good $\delta_\rho$ and $\delta_a$, which are essential to the performance of *PART*.

## Footnotes

[1]MATLAB implementation available from `https://github.com/richardkwo/random-tree-parallel-MCMC`

[2]`https://github.com/wwrechard/weierstrass`

[3]`http://helios.fmi.fi/~lainema/mcmc/`

[4]http://www.csie.ntu.edu.tw/~cjlin/libsvmtools/datasets/binary.html

[5]https://archive.ics.uci.edu/ml/machine-learning-databases/00199

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
