[Supplementary Material]

# Supplementary Materials

**Xiangyu Wang**
Dept. of Statistical Science
Duke University
xw56@stat.duke.edu

**Fangjian Guo**
Dept. of Computer Science
Duke University
guo@cs.duke.edu

**Katherine A. Heller**
Dept. of Statistical Science
Duke University
kheller@stat.duke.edu

**David B. Dunson**
Dept. of Statistical Science
Duke University
dunson@stat.duke.edu

## Appendix A: Schematic illustration of the algorithm

Here we include a schematic illustration of the density aggregation in PART algorithm.

Figure 1: A schematic figure illustrating the *density aggregation* step of the algorithm. Two trees in the left share the same block structure and the aggregated histogram is obtained by block-wise multiplication and renormalization.

## Appendix B: Proof of Theorem 1

Let $\Gamma_{K,\rho}$ be a subset of $\Gamma_K$ defined as $\Gamma_{K,\rho} = \{f_0 \in \Gamma_K | \min_{A_k} \mathbb{E}\mathbf{1}_{A_k} \geq \rho\}$. We prove a more general form of Theorem 1 here.

**Theorem 1.** *For any $\delta > 0$, if the sample size satisfies that $N > \max\{K, \frac{2\log(2K/\delta)}{\rho(1-c_0)^2}\}$, then with probability at least $1 - \delta$, the optimal solution to (8) satisfies that*

$$D_{KL}(f\|\hat{f}_{ML}) \leq \min_{f_0 \in \Gamma_{K,\rho}} D_{KL}(f\|f_0) + C_\rho \sqrt{\frac{K}{N}\log\left(\frac{3eN}{K}\right)\log\left(\frac{8}{\delta}\right)},$$

*where $C_\rho = 48\sqrt{p+1}\max\{\log D, \log \rho^{-1}\}$.*

*Now choosing $\rho = 1/K^{1+1/(2p)}$, the condition becomes $N > 2(1-c_0)^{-2}K^{1+1/(2p)}\log(2K/\delta)$, then with probability at least $1 - \delta$ we have*

$$D_{KL}(f\|\hat{f}_{ML}) \leq (C_1 + 2\log K)K^{-\frac{1}{2p}} + C_2 \max\{\log D, 2\log K\}\sqrt{\frac{K}{N}\log\left(\frac{3eN}{K}\right)\log\left(\frac{8}{\delta}\right)},$$

*where $C_1 = 2\log D + 4pLD$ with $L = \|f'\|_\infty$ and $C_2 = 48\sqrt{p+1}$.*

When multiple densities $f^{(1)}(\theta), \cdots, f^{(m)}(\theta)$ are presented, our goal of imposing the same partition on all functions requires solving a different problem (9). As long as $\hat{\mathbb{E}} \sum_{i=1}^{m} \hat{f}_{ML}^{(i)} \geq \hat{\mathbb{E}} \sum_{i=1}^{m} f_{K,\rho}^{(i)}$ remains true, where $f_{K,\rho}^{(i)} = \arg\min_{f_0 \in \Gamma_{K,\rho}} D_{KL}(f^{(i)} \| f_0)$, the whole proof of Theorem 1 is also valid for (9). Therefore, we have the following Corollary.

**Corollary 1** (*m* copies). *For any $\delta > 0$, if the sample size satisfies that $N > 2(1 - c_0)^{-2} K^{1+1/(2p)} \log(2mK/\delta)$ and $\rho = 1/K^{1+1/(2p)}$, then with probability at least $1 - \delta$, the optimal solution to* (9) *satisfies that*

$$\frac{1}{m} \sum_{i=1}^{m} D_{KL}(f^{(i)} \| \hat{f}_{ML}^{(i)}) \leq (C_1 + 2\log K)K^{-\frac{1}{2p}} + C_2 \max\left\{\log D, 2\log K\right\} \sqrt{\frac{K}{N} \log\left(\frac{3eN}{K}\right) \log\left(\frac{8m}{\delta}\right)}.$$

To prove Theorem 1 we need the following lemmas.

**Lemma 1.** *The optimal solution of* (8) *is also the optimal solution of the following problem,*

$$\hat{f}_{ML} = \arg\max_{f \in \Gamma_K} \frac{1}{N} \sum_{k=1}^{K} n_k \log \hat{\pi}_k, \quad s.t. \ n_k \geq c_0 \rho N, \ |A_k| \geq \rho/D \text{ and } \sum_{k=1}^{K} \hat{\pi}_k |A_k| = 1. \quad (1)$$

*Proof.* We write out the empirical log likelihood

$$\frac{1}{N} \sum_{k=1}^{K} n_k \log \hat{\pi}_k = \sum_{k=1}^{K} \frac{n_k}{N} \log \hat{\pi}_k |A_k| - \sum_{n_k \geq 0} \frac{n_k}{N} \log |A_k|.$$

For any fixed partition $\{A_k\}$, under the constraint that $\sum_{k=1}^{K} \hat{\pi}_k |A_k| = 1$, one can easily see that

$$\hat{\pi}_k |A_k| = \frac{n_k}{N}$$

maximizes the result. □

Next, we show that the optimal approximation

$$f_{K,\rho} = \arg\min_{\hat{f}_{ML} \in \Gamma_{K,\rho}} D_{KL}(f \| \hat{f}_{ML})$$

is a feasible solution to (1) with a high probability.

**Lemma 2.** *Let $f_{K,\rho}$ be the optimal approximation in $\Gamma_{K,\rho}$, then $f_{K,\rho}$ satisfies that $\min_k |A_k| \geq \rho/D$. In addition, with probability at least $1 - K\exp(-(1-c_0)^2 \rho N/2)$, we have $n_k/N \geq c_0 \rho$, i.e., $f_{K,\rho}$ is a feasible soluton of* (1).

*Proof.* Let $n_k$ be the counts of data points on the partition of $f_{K,\rho}$. Notice $f_{K,\rho}$ is a fixed function that does not depend on the data. Therefore, each $n_k$ follows a binomial distribution. Define $P(A_k) = \mathbb{E} \mathbf{1}_{A_k}$. According to the definition of $\Gamma_{K,\rho}$, we have $P(A_k) \geq \rho$. Using the Chernoff's inequality, we have for any $0 < \delta < 1$,

$$P\left(\frac{n_k}{N} \leq (1 - \delta)P(A_k)\right) \leq \exp\left(-\frac{\delta^2 N P(A_k)}{2}\right).$$

Taking $\delta = 1 - c_0$ and union bounds we have

$$P\left(\min_k \frac{n_k}{N} \geq c_0 \rho\right) \geq 1 - K\exp(-(1-c_0)^2 \rho N/2).$$

On the other hand, the following inequality shows the bound on $|A_k|$,

$$|A_k| = \int_{A_k} 1 \geq \int_{A_k} f/D \geq \rho/D.$$

□

Lemma 2 states that with a high probability we have $\hat{\mathbb{E}} \log \hat{f}_{ML} \geq \hat{\mathbb{E}} \log f_{K,\rho}$. This result will be used to prove our main theorem.

Although the actual partition algorithm selects the dimension for partitioning completely at random for each iteration, in the proof we will assume one predetermined order of partition (such as $\{1, 2, 3, \cdots, p, 1, 2, \cdots\}$) just for simplicity. The order of partitioning does not matter as long as every dimension receives sufficient number of partitions. When the selection is randomly taken, with high probability (increasing exponentially with $N$), the number of partitions in each dimension will concentrate around the average. Thus, it suffices to prove the result for the simple $\{1, 2, 3, \cdots, p, 1, 2, \cdots\}$ case.

***Proof of Theorem 1***. The proof consists of two parts, namely (1) bounding the excess loss compared to the optimal approximation $f_{K,\rho}$ in $\Gamma_{K,\rho}$ and (2) bounding the error between the optimal approximation and the true density.

For the first part, using the fact that $\hat{\mathbb{E}} \log f_{K,\rho}(\theta) \leq \hat{\mathbb{E}} \log \hat{f}_{ML}(\theta)$, the excess loss can be expressed as

$$
\begin{aligned}
D_{KL}(f\|\hat{f}_{ML}) - D_{KL}(f\|f_{K,\rho}) &= \mathbb{E} \log f_{K,\rho}(\theta) - \mathbb{E} \log \hat{f}_{ML}(\theta) \\
&= \mathbb{E} \log f_{K,\rho}(\theta) - \hat{\mathbb{E}} \log f_{K,\rho}(\theta) + \hat{\mathbb{E}} \log f_{K,\rho}(\theta) \\
&\quad - \hat{\mathbb{E}} \log \hat{f}_{ML}(\theta) + \hat{\mathbb{E}} \log \hat{f}_{ML}(\theta) - \mathbb{E} \log \hat{f}_{ML}(\theta) \\
&\leq \mathbb{E} \log f_{K,\rho}(\theta) - \hat{\mathbb{E}} \log f_{K,\rho}(\theta) + \hat{\mathbb{E}} \log \hat{f}_{ML}(\theta) - \mathbb{E} \log \hat{f}_{ML}(\theta).
\end{aligned}
$$

Assuming the partitions for $f_{K,\rho}$ and $\hat{f}_{ML}$ are $\{A_k\}$ and $\{\hat{A}_k\}$ respectively, we have

$$
\begin{aligned}
D_{KL}(f\|\hat{f}_{ML}) - D_{KL}(f\|f_{K,\rho}) &= \sum_{k=1}^{K} \log \pi_k \left( \mathbb{E} \mathbf{1}_{A_k} - \hat{\mathbb{E}} \mathbf{1}_{A_k} \right) + \sum_{k=1}^{K} \log \frac{n_k}{N|\hat{A}_k|} \left( \mathbb{E} \mathbf{1}_{\hat{A}_k} - \hat{\mathbb{E}} \mathbf{1}_{\hat{A}_k} \right) \\
&\leq \left( \max_k |\log \pi_k| + \max_k \left| \log \frac{n_k}{N|\hat{A}_k|} \right| \right) \sup_{\{A_k\} \in \mathcal{F}_k} \sum_{k=1}^{K} |\mathbb{E} \mathbf{1}_{A_k} - \hat{\mathbb{E}} \mathbf{1}_{A_k}|. \quad (2)
\end{aligned}
$$

Following a similar argument as Lemma 1, for $f_{K,\rho}$ we can prove that $\pi_k = \int_{A_k} f(\theta) d\theta / |A_k|$, thus we have $\rho_0 \leq \pi_k \leq D$ for any $1 \leq k \leq K$. Similarly for $\frac{n_k}{N|\hat{A}_k|}$ we have $\rho \leq \frac{n_k}{N|\hat{A}_k|} \leq D/\rho$. Therefore, the first term in (2) can be bounded as

$$
\max_k |\log \pi_k| + \max_k \left| \log \frac{n_k}{N|\hat{A}_k|} \right| \leq 3 \max\{\log D, \ \log \rho^{-1}\}.
$$

The second term in (2) is the concentration of the empirical measure over all possible K-rectangular partitions. Using the result from [1], we have the following large deviation inequality. For any $\epsilon \in (0, 1)$, we have

$$
P\left( \sup_{\{A_k\} \in \mathcal{F}_k} \sum_{k=1}^{K} |\mathbb{E} \mathbf{1}_{A_k} - \hat{\mathbb{E}} \mathbf{1}_{A_k}| > \epsilon \right) < 4 \exp\left\{ -\frac{\epsilon^2 N}{2^9} \right\}, \quad (3)
$$

if $N \geq \max\{K, (100 \log 6)/\epsilon^2, 2^9(p+1)K \log(3eN/K)/\epsilon^2\}$. For any $\delta > 0$, taking $\epsilon = 2^9(p+1)K \log(3eN/K)/N \log(4/\delta)$, we have that

$$
\sup_{\{A_k\} \in \mathcal{F}_k} \sum_{k=1}^{K} |\mathbb{E} \mathbf{1}_{A_k} - \hat{\mathbb{E}} \mathbf{1}_{A_k}| \leq 16\sqrt{2(p+1)} \sqrt{\frac{K}{N} \log\left(\frac{3eN}{K}\right) \log\left(\frac{8}{\delta}\right)},
$$

with probability at least $1 - \delta/2$. Define $C_\rho = 48\sqrt{2(p+1)} \max\{\log D, \log \rho^{-1}\}$. When $N > \frac{2 \log(2K/\delta)}{\rho(1-c_0)^2}$, Lemma 2 holds with probability at least $1 - \delta/2$. Taking the union bound, we have

$$
D_{KL}(f\|\hat{f}_{ML}) \leq \min_{f_0 \in \Gamma_{K,\rho}} D_{KL}(f\|f_0) + C_\rho \sqrt{\frac{K}{N} \log\left(\frac{3eN}{K}\right) \log\left(\frac{8}{\delta}\right)} \quad (4)
$$

holds with probability greater than $1 - \delta$.

To prove the second part, we construct one reference density $\tilde{f} \in \Gamma_{K,\rho}$ that gives the error specified in the theorem. According to the argument provided in the paragraph prior to this proof, we assume the dimension that we cut at each iteration follows an order $\{1, 2, \cdots, p, 1, 2, \cdots\}$. We then construct $f_0$ in the following way. At iteration $i$, we check the probability on the whole region.

    i If the probability is greater than $2\rho$, we then cut at the midpoint of the selected dimension. If the resulting two blocks $B_1$ and $B_2$ satisfy that $P_f(B_1) \geq \rho$ and $P_f(B_2) \geq \rho$, we continue to the next iteration. However, if any of them fails to satisfy the condition, we find the minimum-deviated cut that satisfies the probability requirement.

    ii If the probability on the whole region is less than $2\rho$, we stop cutting on this region and move to the next region for the current iteration.

It is easy to show that as long as $\rho \leq 1/(2K)$, the above procedure is able to yield a $K$-block partition $\{\tilde{A}_k\}$ before termination. Finally, the reference density $\tilde{f}$ is defined as

$$\tilde{f}(\theta) = \sum_{k=1}^{K} \frac{\int_{\tilde{A}_k} f(\theta)d\theta}{|\tilde{A}_k|} 1_{\tilde{A}_k}(\theta). \tag{5}$$

The construction procedure ensures the following property for $\tilde{f} \in \Gamma_{K,\rho}$. Assuming $K \in [2^d, 2^{d+1})$ for some $d > 0$, then each $\tilde{A}_k$ must fall into either of the following two categories (could be both),

1. $\rho \leq P_f(\tilde{A}_k) \leq 2\rho$,

2. $P_f(\tilde{A}_k) \geq \rho$ and the longest edge of cube $\tilde{A}_k$ must be less than $2^{-\lfloor d/p \rfloor} \leq 2^{-d/p+1} \leq 4k^{-1/p}$.

We use $I_1$ and $I_2$ to denote the two different collections of sets. Now for any $b_0 > 0$, let $B = \{f < b_0\} \cup \{\tilde{f} < b_0\}$. We divide the KL-divergence between $f$ and $\tilde{f}$ into three regions and bound them accordingly.

$$D_{KL}(f\|\tilde{f}) = \int_\Omega f \log \frac{f}{\tilde{f}} = \int_B f \log \frac{f}{\tilde{f}} + \int_{B^c \cap \left( \bigcup I_1 \right)} f \log \frac{f}{\tilde{f}} + \int_{B^c \cap \left( \bigcup I_2 \right)} f \log \frac{f}{\tilde{f}}$$
$$= M_1 + M_2 + M_3.$$

We first look at $M_1$. Because $\tilde{f}$ is a block-valued function, $\{\tilde{f} < b_0\}$ must be the union of all the $\tilde{A}_k$ that satisfies $\int_{\tilde{A}_k} f(\theta)d\theta \leq b_0|\tilde{A}_k|$. Therefore, we have

$$\int_{\tilde{f} < b_0} f(\theta)d\theta = \sum_{\tilde{A}_k: \ \int_{\tilde{A}_k} f(\theta)d\theta \leq b_0|\tilde{A}_k|} \int_{\tilde{A}_k} f(\theta)d\theta \leq \sum_{\tilde{A}_k: \ \int_{\tilde{A}_k} f(\theta)d\theta \leq b_0|\tilde{A}_k|} b_0|\tilde{A}_k| \leq b_0.$$

Therefore, we have

$$\int_B f(\theta)d\theta \leq \int_{\tilde{f} < b_0} f(\theta)d\theta + \int_{f < b_0} f(\theta)d\theta = b_0 + b_0|\Omega| = 2b_0.$$

Because $\tilde{f} \geq \min_k P(A_k)/|A_k| \geq P(A_k) \geq \rho$, we have

$$M_1 = \int_B f \log \frac{f}{\tilde{f}} \leq \int_B f(\theta) \log \frac{b_0}{\rho} \leq 2b_0 \big| \log \frac{b_0}{\rho} \big|.$$

Next, we look at $M_2$. It is clear that

$$\int_{B^c \cap \left( \bigcup I_1 \right)} f(\theta)d\theta \leq \int_{\left( \bigcup I_1 \right)} f(\theta)d\theta \leq card(I_1)2\rho,$$

and hence we have

$$M_2 = \int_{B^c \cap \left(\bigcup I_1\right)} f \log \frac{f}{\tilde{f}} \leq \int_{B^c \cap \left(\bigcup I_1\right)} f(\theta) \log \frac{D}{b_0} \leq card(I_1) 2\rho \big| \log \frac{D}{b_0} \big| \leq 2K\rho \big| \log \frac{D}{b_0} \big|.$$

Now for $M_3$, we first use the inequality that $\log x \leq x - 1$ for any $x > 0$,

$$M_3 = \int_{B^c \cap \left(\bigcup I_2\right)} f \log \frac{f}{\tilde{f}} \leq \int_{B^c \cap \left(\bigcup I_2\right)} f\left(\frac{f}{\tilde{f}} - 1\right) \leq \int_{B^c \cap \left(\bigcup I_2\right)} \frac{f}{\tilde{f}}(f - \tilde{f}) \leq \frac{D}{b_0} \int_{\bigcup I_2} |f - \tilde{f}|.$$

Using the mean value theorem for integration, we have $\int_{\tilde{A}_k} f(\theta)/|\tilde{A}_k| = f(\theta_0)$ for some $\theta_0 \in \tilde{A}_k$. Also because $f \in C^1(\Omega)$, $\|f'\|_\infty$ is bounded, i.e., there exists some constant $L$ such that $|f(x_1) - f(x_2)| \leq L \sum_{j=1}^{p} |x_{1j} - x_{2j}|$. Therefore, we have

$$\int_{\bigcup I_2} |f - \tilde{f}| = \sum_{\tilde{A}_k \in I_2} \int_{\tilde{A}_k} |f(\theta) - \tilde{f}(\theta)| = \sum_{\tilde{A}_k \in I_2} \int_{\tilde{A}_k} |f(\theta) - f(\theta_0)| \leq \sum_{\tilde{A}_k \in I_2} 4pLk^{-\frac{1}{p}} |\tilde{A}_k| \leq 4pLk^{-\frac{1}{p}},$$

and thus

$$M_3 \leq \frac{4pLD}{b_0} K^{-\frac{1}{p}}.$$

Putting all pieces together we have

$$D_{KL}(f\|\tilde{f}) \leq 2b_0 \big| \log \frac{b_0}{\rho} \big| + 2K\rho \big| \log \frac{D}{b_0} \big| + \frac{4pLD}{b_0} K^{-\frac{1}{p}}.$$

Now taking $b_0 = K^{-1/(2p)}$ and $\rho = K^{-1-1/(2p)}$, we have

$$D_{KL}(f\|\tilde{f}) \leq (\log K)K^{-\frac{1}{2p}} + 2(\log D + \frac{\log K}{2p})K^{-\frac{1}{2p}} + 4pLDK^{-\frac{1}{2p}}$$

$$\leq (2\log K + 2\log D + 4pLD)K^{-\frac{1}{2p}}.$$

Now defining $C_1 = 2\log D + 4pLD$ and $C_2 = 48\sqrt{p+1}$ and combining with (4), we have

$$D_{KL}(f\|\hat{f}_{ML}) \leq (C_1 + 2\log K)K^{-\frac{1}{2p}} + C_2 \max\big\{\log D, 2\log K\big\} \sqrt{\frac{K}{N} \log\left(\frac{3eN}{K}\right) \log\left(\frac{8}{\delta}\right)}.$$

$\square$

## Appendix C: Proof of Theorem 2

The KD-tree $\hat{f}_{KD}$ always cuts at the empirical median for different dimensions, aiming to approximate the true density by equal probability partitioning. For $\hat{f}_{KD}$ we have the following result.

**Theorem 2.** *For any* $\varepsilon > 0$*, define* $r_\varepsilon = \log_2\left(1 + \frac{1}{2+3L/\varepsilon}\right)$*. For any* $\delta > 0$*, if* $N > 32e^2(\log K)^2 K \log(2K/\delta)$*, then with probability at least* $1 - \delta$*, we have*

$$\|\hat{f}_{KD} - f_{KD}\|_1 \leq \varepsilon + pLK^{-\frac{r_\varepsilon}{p}} + 4e\log K \sqrt{\frac{2K}{N} \log\left(\frac{2K}{\delta}\right)}.$$

*If the function is further lower bounded by some constant* $b_0 > 0$*, we can then obtain an upper bound on the KL-divergence. Define* $r_{b_0} = \log_2\left(1 + \frac{1}{2+3L/b_0}\right)$ *and we have*

$$D_{KL}(f\|\hat{f}_{KD}) \leq \frac{pLD}{b_0} K^{-\frac{r_{b_0}}{p}} + 8e\log K \sqrt{\frac{2K}{N} \log\left(\frac{2K}{\delta}\right)}.$$

When there are multiple functions and the median partition is performed on pooled data, the partition might not happen at the empirical median on each subset. However, as long as the partition quantiles are upper and lower bounded by $\alpha$ and $1 - \alpha$ for some $\alpha \in [1/2, 1)$, we can establish similar theory as Theorem 2.

**Corollary 2.** *Assume we instead partition at different quantiles that are upper and lower bounded by $\alpha$ and $1 - \alpha$ for some $\alpha \in [1/2, 1)$. Define $r_\varepsilon = \log_2 \left( 1 + \frac{(1-\alpha)}{2\alpha + 3L/\varepsilon + 1} \right)$ and $r_{b_0} = \log_2 \left( 1 + \frac{(1-\alpha)}{2\alpha + 3L/b_0 + 1} \right)$. For any $\delta > 0$, if $N > \frac{12e^2}{(1-\alpha)^2} K (\log K)^2 \log(K/\delta)$, then with probability at least $1 - \delta$ we have*

$$\|\hat{f}_{KD} - f_{KD}\|_1 \le \varepsilon + pLK^{-\frac{r_\varepsilon}{p}} + \frac{2e \log K}{1 - \alpha} K^{1 - \log_2 \alpha^{-1}} \sqrt{\frac{3K}{N} \log \left( \frac{2K}{\delta} \right)}$$

*and if the function is lower bounded by $b_0$, then we have*

$$D_{KL}(f \| \hat{f}_{KD}) \le \frac{pLD}{b_0} K^{-\frac{r_{b_0}}{p}} + \frac{4e \log K}{1 - \alpha} K^{1 - \log_2 \alpha^{-1}} \sqrt{\frac{3K}{N} \log \left( \frac{2K}{\delta} \right)}.$$

Following the same argument for Theorem 1, we will prove Theorem 2 by assuming a predetermined order of partition (such as $\{1, 2, 3, \cdots, p, 1, 2, \cdots\}$) for simplicity, though in the actual precedure, the dimensions are selected completely at random. We need the following two lemmas to prove Theorem 2. Let $f_{KD}$ have the same partition as $\hat{f}_{KD}$ but with function value replaced by the true probability on each region divided by the area, i.e., $f_{KD} = \sum_{A_k} \frac{\int_{A_k} f(\theta) d\theta}{|A_k|} \mathbb{1}_{A_k}(\theta)$.

**Lemma 3.** *With $f_{KD}$ defined above, for any $\delta > 0$, if $N > 32e^2 (\log K)^2 K \log(K/\delta)$, then with probability at least $1 - \delta$, we have*

$$\|\hat{f}_{KD} - f_{KD}\|_1 \le 4e \log K \sqrt{\frac{2K}{N} \log \left( \frac{K}{\delta} \right)}$$

*and*

$$D_{KL}(f \| \hat{f}_{KD}) \le D_{KL}(f \| f_{KD}) + 8e \log K \sqrt{\frac{2K}{N} \log \left( \frac{K}{\delta} \right)}.$$

*If we instead partition at some different quantiles, which are upper bounded by $\alpha$ and lower bounded by $1 - \alpha$ for some $\alpha \in [1/2, 1)$, then for any $\delta > 0$, if $N > \frac{12e^2}{(1-\alpha)^2} K (\log K)^2 \log(K/\delta)$, with probability at least $1 - \delta$ we have*

$$\|\hat{f}_{KD} - f_{KD}\|_1 \le \frac{2e \log K}{1 - \alpha} K^{1 - \log_2 \alpha^{-1}} \sqrt{\frac{3K}{N} \log \left( \frac{K}{\delta} \right)},$$

*and*

$$D_{KL}(f \| \hat{f}_{KD}) \le D_{KL}(f \| f_{KD}) + \frac{4e \log K}{1 - \alpha} K^{1 - \log_2 \alpha^{-1}} \sqrt{\frac{3K}{N} \log \left( \frac{K}{\delta} \right)}.$$

*Proof.* The proof is based on how close the data median is to the true median. Suppose there are $N_i$ points in the current region, condition on this region, and partition it into two regions $\hat{A}_1$ and $\hat{A}_2$ by cutting at the data median point $\hat{M}_i$. Denote the true median by $M_i$ and two anchor points $M_i - \epsilon_1$, $M_i + \epsilon_2$ such that $P(X \le M_i - \epsilon_1) = 1/2 - t$ and $P(X \le M_i + \epsilon_2) = 1/2 + t$ for some $0 < t < 1/2$. By Chernoff's inequality we have

$$P(\hat{M}_i \le M_i - \epsilon_1) \le \exp \left\{ -\frac{t^2 N_i}{1 + 2t} \right\}$$

and
$$P(\hat{M}_i \geq M_i + \epsilon_2) \leq \exp\left\{-\frac{t^2 N_i}{1+2t}\right\}.$$

The above two inequalities indicate that with high probability $\hat{M}_i$ is within the interval $(M_i - \epsilon_1, M_i + \epsilon_2)$. Therefore, the probabilities on $A_1$ and $A_2$ also satisfy that

$$P\left(\left|\mathbb{E}\mathbf{1}_{A_i} - \frac{1}{2}\right| \geq t\right) \leq \exp\left\{-\frac{t^2 N_i}{1+2t}\right\}. \tag{6}$$

Now consider the $K$ regions of $\hat{f}_{KD}$ and $f_{KD}$. Each partition will bring an error of at most $1/2 + t$ to the estimation of the region probability. Therefore, assuming $K \in [2^d + 1, 2^{d+1})$ we have for each region $A_k$ ($A_k$ is a random variable) that

$$\left(\frac{1}{2} - t\right)^d \leq \int_{A_k} f(\theta)d\theta \leq \left(\frac{1}{2} + t\right)^d$$

if $n_k/N = 1/2^d$, or

$$\left(\frac{1}{2} - t\right)^{d+1} \leq \int_{A_k} f(\theta)d\theta \leq \left(\frac{1}{2} + t\right)^{d+1},$$

if $n_k/N = 1/2^{d+1}$. Notice that for all iterations before the current partition, we always have $N_i \geq N/K$. Therefore the probability is guaranteed to be greater than $1 - K \exp\left\{-\frac{t^2 N/K}{1+2t}\right\}$.
The above result indicates that

$$\max_{A_k}\left|\int_{A_k} f_{KD}(\theta) - \int_{A_k} \hat{f}_{KD}(\theta)\right| \leq \max\left\{\left(\frac{1}{2} + t\right)^{d+1} - \left(\frac{1}{2}\right)^{d+1}, -\left(\frac{1}{2} - t\right)^{d+1} + \left(\frac{1}{2}\right)^{d+1}\right\}$$

$$= \left(\frac{1}{2}\right)^{d+1} \max\left\{(1+2t)^{d+1} - 1, 1 - (1-2t)^{d+1}\right\}$$

$$= \left(\frac{1}{2}\right)^{d+1}\left((d+1)(1+2\tilde{t})^d 2t\right),$$

where $\tilde{t} \in (0, t)$. So if $t < 1/(2d)$, then $(1+2\tilde{t})^d \leq (1+1/d)^d < e$ and we have

$$\max_{A_k}\left|\int_{A_k} f_{KD}(\theta) - \int_{A_k} \hat{f}_{KD}(\theta)\right| \leq 2(d+1)e\left(\frac{1}{2}\right)^{d+1} t \leq \frac{4et\log K}{K}. \tag{7}$$

This result implies that the total variation distance satisfies that

$$\|\hat{f}_{KD} - f_{KD}\|_1 = \sum_k \int_{A_k} |\hat{f}_{KD}(\theta) - f_{KD}(\theta)| = \sum_k \left|\int_{A_k} \hat{f}_{KD}(\theta) - \int_{A_k} f_{KD}(\theta)\right| \leq 4et\log K.$$

Similarly, one can also prove that

$$\max_{A_k}\left|\frac{\int_{A_k} f_{KD}(\theta)}{\int_{A_k} \hat{f}_{KD}(\theta)} - 1\right| \leq \max_{A_k}\left|\frac{\int_{A_k} f_{KD}(\theta) - \int_{A_k} \hat{f}_{KD}(\theta)}{\int_{A_k} \hat{f}_{KD}(\theta)}\right| \leq 4et\log K.$$

Denote $\int_{A_k} f(\theta) = \int_{A_k} f_{KD}(\theta)$ by $P(A_k)$ and $\int_{A_K} \hat{f}_{KD} f(\theta)$ by $\hat{P}(A_k)$. The KL-divergence can then be computed as

$$D_{KL}(f\|\hat{f}_{KD}) - D_{KL}(f\|f_{KD}) = \sum_{A_k} \int_{A_k} f(\theta)(\log f_{KD}(\theta) - \log \hat{f}_{KD}(\theta))$$

$$= \sum_{A_k} \int_{A_k} f(\theta)\left(\log \int_{A_k} f(\theta) - \log \int_{A_k} \hat{f}_{KD}(\theta)\right)$$

$$= \sum_{A_k} P(A_k)\left(\log \frac{P(A_k)}{\hat{P}(A_k)}\right) \leq \sum_{A_k} P(A_k)\left(\frac{P(A_k)}{\hat{P}(A_k)} - 1\right)$$

$$= \sum_{A_k} \frac{P(A_k)}{\hat{P}(A_k)}\left(P(A_k) - \hat{P}(A_k)\right)$$

$$\leq (1 + 4et\log K)4et\log K \leq 8et\log K,$$

as long as $t < \min\left\{\frac{1}{4e\log K}, \frac{1}{2\log_2 K}\right\}$, with probability at least $1 - K\exp\left\{-\frac{t^2 N}{2K}\right\}$. Consequently, for any $\delta > 0$, if $N > 32e^2 K(\log K)^2 \log(K/\delta)$, then with probability at least $1 - \delta$, we have

$$\|\hat{f}_{KD} - f_{KD}\|_1 \leq 4e\log K\sqrt{\frac{2K}{N}\log\left(\frac{K}{\delta}\right)}$$

and

$$D_{KL}(f\|\hat{f}_{KD}) \leq D_{KL}(f\|f_{KD}) + 8e\log K\sqrt{\frac{2K}{N}\log\left(\frac{K}{\delta}\right)}.$$

When the partition occurs at a different quantile, which is assumed to be $\alpha_i$ for iteration $i$, we have

$$\prod_{i=1}^{d+1}(\alpha_i' - t) \leq \int_{A_k} f(\theta)d\theta \leq \prod_{i=1}^{d+1}(\alpha_i' + t),$$

where $\alpha_i' = \alpha_i$ if $A_k$ takes the region containing smaller data values and $\alpha_i' = 1 - \alpha_i$ if $A_k$ takes the other half. First, (6) can be updated as

$$P\left(\left|\mathbb{E}1_{A_i} - \alpha_i'\right| \geq t\right) \leq \exp\left\{-\frac{t^2 N_i}{3\alpha}\right\} \leq \exp\left\{-\frac{t^2 N_i}{3}\right\}, \tag{8}$$

if $t < 1 - \alpha$. Then we can bound the difference between $\int_{A_k} f(\theta)$ and $\int_{A_k} \hat{f}_{KD}(\theta)$ as

$$\max_{A_k}\left|\int_{A_k} f_{KD}(\theta) - \int_{A_k}\hat{f}_{KD}(\theta)\right| \leq \max\left\{\prod_{i=1}^{d+1}(\alpha_i' + t) - \prod_{i=1}^{d+1}\alpha_i', \; -\prod_{i=1}^{d+1}(\alpha_i' - t) + \prod_{i=1}^{d+1}\alpha_i'\right\}$$

$$\leq \max\left[\prod_{i=1}^{d+1}\alpha_i'\left\{\prod_{i=1}^{d+1}\left(1 + \frac{t}{\alpha_i'}\right) - 1\right\}, \; \prod_{i=1}^{d+1}\alpha_i'\left\{1 - \prod_{i=1}^{d+1}\left(1 - \frac{t}{\alpha_i'}\right)\right\}\right]$$

$$\leq \prod_{i=1}^{d+1}\alpha_i' \cdot (d+1)\left(1 + \frac{\tilde{t}}{1-\alpha}\right)^d \frac{t}{1-\alpha},$$

where $\tilde{t} \in (0, t)$. Thus if $t < (1-\alpha)/d$, then we have

$$\max_{A_k}\left|\int_{A_k} f_{KD}(\theta) - \int_{A_k}\hat{f}_{KD}(\theta)\right| \leq \frac{e(d+1)t\alpha^{d+1}}{1-\alpha} \leq \frac{2et\log K}{(1-\alpha)K^{\log_2 \alpha^{-1}}},$$

and

$$\max_{A_k}\left|\frac{\int_{A_k} f_{KD}(\theta)}{\int_{A_k}\hat{f}_{KD}(\theta)} - 1\right| \leq \frac{2et\log K}{1-\alpha}.$$

The total variation distance follows

$$\|\hat{f}_{KD} - f_{KD}\|_1 \leq K \cdot \frac{2et\log K}{(1-\alpha)K^{\log_2 \alpha^{-1}}} = \frac{2et\log K}{1-\alpha}K^{1-\log_2 \alpha^{-1}},$$

and the KL-divergence follows

$$D_{KL}(f\|\hat{f}_{KD}) - D_{KL}(f\|f_{KD}) \leq \left(1 + \frac{2et\log K}{1-\alpha}\right)\frac{2et\log K}{1-\alpha}K^{1-\log_2 \alpha^{-1}} \leq \frac{4et\log K}{1-\alpha}K^{1-\log_2 \alpha^{-1}},$$

if $t < 1/(2e(1-\alpha)\log K)$. Consequently, for any $\delta > 0$, if $N > \frac{12e^2}{(1-\alpha)^2}K(\log K)^2 \log(K/\delta)$, then with probability at least $1 - \delta$, we have

$$\|\hat{f}_{KD} - f_{KD}\|_1 \leq \frac{2e\log K}{1-\alpha}K^{1-\log_2 \alpha^{-1}}\sqrt{\frac{3K}{N}\log\left(\frac{K}{\delta}\right)},$$

and

$$D_{KL}(f\|\hat{f}_{KD}) \leq D_{KL}(f\|f_{KD}) + \frac{4e\log K}{1-\alpha}K^{1-\log_2 \alpha^{-1}}\sqrt{\frac{3K}{N}\log\left(\frac{K}{\delta}\right)}.$$

$\square$

Our next result is to bound the distance between $f_{KD}$ and the true density $f$. Again, the proof depends on the control of the smallest value of $f$ and the longest edge of every block. One issue now is that each partition might not happen at the midpoint, but it should not deviate from the midpoint too much given the bound on the $f'$, i.e., we have the following proposition.

**Proposition 1.** *Assume we aim to partition an edge of length $h$ (on dimension $q$) of a rectangular region $A$, which has a probability of $P$ and an area of $|A|$. We distinguish the resulting two regions as the left and the right region and the corresponding edges (on dimension $q$) as $h_{left}$ and $h_{right}$ (i.e., $h_{left} + h_{right} = h$). Suppose the partition ensures that the left region has probability of $\gamma P$, where $\gamma \geq 1/2$. If $\|f'\|_\infty \leq L$, then the longer edge $h^* = \max\{h_{left}, h_{right}\}$ satisfies that*

$$\frac{h^*}{h} \leq 1 - \frac{1 - \gamma}{1 + Lh\frac{|A|}{P}}.$$

*Proof.* It suffices to bound $h_{\text{left}}$ as $\gamma > 1 - \gamma$. Let $g(t) = \int_{x:(t,x)\in A} f(t,x)$, where $t$ represents the variable of dimension $q$ and $x$ stands for the other dimensions. We then have $\int_{t:h} g(t) = P$, $\int_{x:(t,x)\in A} 1dx = |A|/h$ and

$$|g(t_1) - g(t_2)| \leq \int_{x:(t,x)} (f(t_1,x) - f(t_2,x)) \leq L|t_1 - t_2||A|/h.$$

Therefore, using the mean value theorem for the integration, we know that

$$\left| \frac{\int_{t:h_{\text{left}}} g(t)}{h_{\text{left}}} - \frac{\int_{t:h_{\text{right}}} g(t)}{h_{\text{right}}} \right| \leq L|A|,$$

which implies that

$$\left| \frac{\gamma h}{h_{\text{left}}} - \frac{(1-\gamma)h}{h_{\text{right}}} \right| \leq \frac{L|A|h}{P}.$$

Now if we solve the following inequality

$$|\gamma/a - (1-\gamma)/b| \leq c \quad \text{and} \quad a + b = 1, a \geq 0, b \geq 0,$$

with some simple algebra we can get

$$a \leq 1 - \frac{1-\gamma}{1+c}.$$

Plug in the corresponding value, we have

$$\frac{h_{\text{left}}}{h} \leq 1 - \frac{1-\gamma}{1 + Lh\frac{|A|}{P}}.$$

$\square$

With Proposition 1, we can now obtain the upper bound for $\|f - f_{KD}\|_1$ and $D_{KL}(f\|f_{KD})$.

**Lemma 4.** *For any $\varepsilon > 0$, define $r_\varepsilon = \log_2\left(1 + \frac{1}{2+3L/\varepsilon}\right)$. If $N \geq 72K \log(K/\delta)$ for any $\delta > 0$, then with probability at least $1 - \delta$, we have*

$$\|f - f_{KD}\|_1 \leq \varepsilon + pLK^{-\frac{r_\varepsilon}{p}}.$$

*If the $f$ is further lower bounded by some $b_0 > 0$, the KL-divergence can be bounded as*

$$D_{KL}(f\|f_{KD}) \leq \frac{pLD}{b_0} K^{-\frac{r_{b_0}}{p}},$$

*where $r_{b_0} = \log_2\left(1 + \frac{1}{2+3L/b_0}\right)$.*

*Now suppose we instead partition at different quantiles, upper and lower bounded by $\alpha$ and $1 - \alpha$ for some $\alpha \in (1/2, 1)$. For any $\delta > 0$, if $N \geq \frac{27}{(1-\alpha)^2}K \log(K/\delta)$ then the above two bounds hold with different $r_\varepsilon$ and $r_{b_0}$ as*

$$r_\varepsilon = \log_2\left(1 + \frac{(1-\alpha)}{2\alpha + 3L/\varepsilon + 1}\right) \quad \text{and} \quad r_{b_0} = \log_2\left(1 + \frac{(1-\alpha)}{2\alpha + 3L/b_0 + 1}\right).$$

*Proof.* The proof for the total variation distance follows similarly as Theorem 1. For any $\varepsilon > 0$, we consider $B = \{f_{KD} < \varepsilon/2\}$. We then partition the total variation distance formula into two parts

$$\|f_{KD} - f\|_1 = \int_B |f_{KD} - f| + \int_{B^c} |f_{KD} - f| = M_1 + M_2.$$

It is straightforward to bound $M_1$. $B$ is a union of $A_k$'s which satisfies $\int_{A_k} f(\theta) \leq \varepsilon |A_k|/2$. Therefore,

$$M_1 \leq \int_B f + \int_B f_{KD} = \sum_{A_k: \cup A_k = B} 2 \int_{A_k} f(\theta) \leq \varepsilon.$$

Now for $M_2$, the usual analysis shows that our result depends on the longest edge of each block, i.e.,

$$M_2 = \int_{B^c} |f_{KD} - f| = \sum_{A_k: \cup A_k = B^c} \int_{A_k} |f_{KD} - f| \leq \sum_{A_k: \cup A_k = B^c} pLh_k^*|A_k| = pL|B^c| \max_{A_k} h_k^* \leq pL \max_{A_k} h_k^*,$$

where $h_k^*$ is the longest edge of each block contained in $B^c$. Now using Proposition 1, we know for iteration $i$ the partitioned edge at each block follows

$$h_i \leq \left(1 - \frac{1-\gamma}{1 + Lh\frac{|A|}{P}}\right) h_{i-1} \leq \left(1 - \frac{1-\gamma}{1 + L/\varepsilon}\right) h_{i-1}.$$

When $K \in (2^d, 2^{d+1}]$, each dimension receives $\lfloor d/p \rfloor$ stages of partitioning; therefore, we have for each block, the longest edge satisfies that

$$h^* \leq \left(1 - \frac{1-\gamma}{1 + L/\varepsilon}\right)^{\frac{\log_2 K}{p}} \leq K^{-\frac{r_\varepsilon}{p}},$$

where $r_\varepsilon = \log_2 \left(1 + \frac{(1-\gamma)}{\gamma + L/\varepsilon}\right)$. This implies

$$M_2 \leq pLK^{-\frac{r_\varepsilon}{p}} \quad \text{and} \quad \|f_{KD} - f\|_1 \leq \varepsilon + pLK^{-\frac{r_\varepsilon}{p}}.$$

Now, according to (6), we know with probability at least $1 - K\exp\{-t^2 N/(2K)\}$,

$$\gamma \leq \frac{1}{2} + t.$$

Taking $t = 1/6$, we get

$$r_\varepsilon = \log_2 \left(1 + \frac{1}{2 + 3L/\varepsilon}\right),$$

with probability at least $1 - K\exp\{-N/(72K)\}$. So if $N > 72K \log(K/\delta)$, then the probability is at least $1 - \delta$. For the case when $\gamma = \alpha + t$, we choose $t = (1-\alpha)/3$, then

$$r_\varepsilon = \log_2 \left(1 + \frac{(1-\alpha)}{2\alpha + 3L/\varepsilon + 1}\right)$$

with probability at least $1 - \delta$ if $N > \frac{27}{(1-\alpha)^2} K \log(K/\delta)$.

For KL-divergence, if $f$ is lower bounded by some constant $b_0 > 0$, then we know that

$$D_{KL}(f\|f_{KD}) = \int_\Omega f(\theta) \log \frac{f(\theta)}{f_{KD}(\theta)} \leq \int_\Omega f(\theta) \left(\frac{f(\theta)}{f_{KD}(\theta)} - 1\right)$$

$$\leq \max_\theta \frac{f(\theta)}{f_{KD}(\theta)} \int_\Omega |f(\theta) - f_{KD}(\theta)| \leq \frac{D}{b_0} \|f - f_{KD}\|_1.$$

Because $f$ and $f_{KD}$ are both lower bounded by $b_0$, we can follow the proof for $\|f - f_{KD}\|_1$ with $\varepsilon = b_0$ and ignore $M_1$. Thus we have

$$D_{KL}(f\|f_{KD}) \leq \frac{pLD}{b_0} K^{-\frac{r_{b_0}}{p}},$$

where $r_{b_0} = \log_2\left(1 + \frac{(1-\gamma)}{\gamma + L/b_0}\right)$. Similarly, if we take $\gamma = 2/3$ and $N \geq 72K\log(K/\delta)$, then with probability at least $1 - \delta$, we have

$$r_{b_0} = \log_2\left(1 + \frac{1}{2 + 3L/b_0}\right).$$

If we take $\gamma = (2\alpha + 1)/3$ and $N > \frac{27}{(1-\alpha)^2}K\log(K/\delta)$, then with probability at least $1 - \delta$, we have

$$r_{b_0} = \log_2\left(1 + \frac{(1-\alpha)}{2\alpha + 3L/b_0 + 1}\right).$$

$\square$

Theorem 2 and Corollary 2 follow directly from Lemma 3 and 4.

***Proof of Theorem 2 and Corollary 2.*** For any $\varepsilon > 0$, define $r_\varepsilon$ and $r_{b_0}$ as in Lemma 4. Thus, for any $\delta > 0$, if $N > 32e^2(\log K)^2 K\log\frac{2K}{\delta}$, then with probability $1 - \delta/2$ we have

$$\|\hat{f}_{KD} - f_{KD}\|_1 \leq 4e\log K\sqrt{\frac{2K}{N}\log\left(\frac{2K}{\delta}\right)}$$

and

$$D_{KL}(f\|\hat{f}_{KD}) \leq D_{KL}(f\|f_{KD}) + 8e\log K\sqrt{\frac{2K}{N}\log\left(\frac{2K}{\delta}\right)}.$$

Also, with probability $1 - \delta/2$ we have

$$\|f - f_{KD}\|_1 \leq \varepsilon + pLK^{-\frac{r_\varepsilon}{p}},$$

and

$$D_{KL}(f\|f_{KD}) \leq \frac{pLD}{b_0}K^{-\frac{r_{b_0}}{p}}.$$

Putting the two equations together we have

$$\|\hat{f}_{KD} - f_{KD}\|_1 \leq \varepsilon + pLK^{-\frac{r_\varepsilon}{p}} + 4e\log K\sqrt{\frac{2K}{N}\log\left(\frac{2K}{\delta}\right)},$$

and

$$D_{KL}(f\|\hat{f}_{KD}) \leq \frac{pLD}{b_0}K^{-\frac{r_{b_0}}{p}} + 8e\log K\sqrt{\frac{2K}{N}\log\left(\frac{2K}{\delta}\right)}.$$

Using the same argument on random quantiles, if $N > \frac{12e^2}{(1-\alpha)^2}K(\log K)^2\log(2K/\delta)$, then with probability at least $1 - \delta$ we have

$$\|\hat{f}_{KD} - f_{KD}\|_1 \leq \varepsilon + pLK^{-\frac{r_\varepsilon}{p}} + \frac{2e\log K}{1-\alpha}K^{1-\log_2\alpha^{-1}}\sqrt{\frac{3K}{N}\log\left(\frac{2K}{\delta}\right)}$$

and

$$D_{KL}(f\|\hat{f}_{KD}) \leq \frac{pLD}{b_0}K^{-\frac{r_{b_0}}{p}} + \frac{4e\log K}{1-\alpha}K^{1-\log_2\alpha^{-1}}\sqrt{\frac{3K}{N}\log\left(\frac{2K}{\delta}\right)},$$

where $r_\varepsilon$ and $r_{b_0}$ are defined as

$$r_\varepsilon = \log_2\left(1 + \frac{(1-\alpha)}{2\alpha + 3L/\varepsilon + 1}\right) \quad \text{and} \quad r_{b_0} = \log_2\left(1 + \frac{(1-\alpha)}{2\alpha + 3L/b_0 + 1}\right).$$

$\square$

## Appendix E: Proof of Theorem 3 and 4

**Lemma 5.** *Assume* $\|f\|_\infty \le D$*. Under the same condition as Theorems 1 and 2, if*

$$\sqrt{N} \ge 32c_0^{-1}\sqrt{2(p+1)}K^{\frac{3}{2}+\frac{1}{2p}}\sqrt{\log\left(\frac{3eN}{K}\right)\log\left(\frac{8}{\delta}\right)},$$

*then we have* $\|\hat{f}_{ML}\|_\infty \le 2D$ *and if*

$$N > 128e^2 K(\log K)^2 \log(K/\delta),$$

*we have* $\|\hat{f}_{KD}\|_\infty \le 2D$.

*Proof.* Assume $\|f\|_\infty \le D$. We want to bound $\|\hat{f}_{ML}\|_\infty$ and $\|\hat{f}_{KD}\|_\infty$. Define

$$\tilde{f} = \sum_{A_K} \frac{\int_{A_k} f(\theta)d\theta}{|A_k|} 1_{A_k}(\theta),$$

which clearly satisfies $\tilde{f} \le D$. Notice that if there exists some $\epsilon$ such that

$$\max_{A_k} |P(A_k) - \hat{P}(A_k)| \le \epsilon,$$

where $P(A_k) = \mathbb{E}\,1_{A_k}$ and $\hat{P}(A_k) = \hat{\mathbb{E}}\,1_{A_k}$, then we have

$$\|\tilde{f} - \hat{f}\|_\infty = \max_{A_k}\left|\frac{P(A_k)}{|A_k|} - \frac{\hat{P}(A_k)}{|A_k|}\right| = \max_{A_k}\frac{1}{|A_k|}|P(A_k) - \hat{P}(A_k)|$$

$$\le \max_{A_k}\frac{\epsilon\tilde{f}(\theta)}{P(A_k)} \le \max_{A_k}\frac{\epsilon D}{\hat{P}(A_k) - \epsilon}.$$

Now if we can pick $\epsilon = \min_{A_k}\hat{P}(A_k)/2$, then the upper bound becomes $2D$. We deduce the corresponding condition for ML-cut and KD-cut respectively. For maximum likelihood partition, plug in $\epsilon = K^{-1-1/(2p)}/2$ into (3). Under the condition of Theorem 1, if

$$\sqrt{N} \ge 32c_0^{-1}\sqrt{2(p+1)}K^{\frac{3}{2}+\frac{1}{2p}}\sqrt{\log\left(\frac{3eN}{K}\right)\log\left(\frac{8}{\delta}\right)},$$

then with probability at least $1 - \delta/2$, we have

$$\|\hat{f}_{ML}\|_\infty \le 2D.$$

For median partition, choose $\epsilon = K^{-1}/2$ and apply (7). Under the condition of Theorem 2, if

$$N > 128e^2 K(\log K)^2 \log(K/\delta),$$

then with probability at least $1 - \delta$, we have

$$\|\hat{f}_{KD}\|_\infty \le 2D.$$

$\square$

***Proof of Theorem 3.*** Assume the average total variation distance between $\hat{f}^{(i)}$ and $f^{(i)}$ is $\varepsilon$. It can be calculated directly as

$$\int\left|\prod_{i\in I} f^{(i)}(\theta) - \prod_{i\in I}\hat{f}^{(i)}(\theta)\right|d\theta \le \sum_{i=1}^{m}\int|f^{(i)}(\theta) - \hat{f}^{(i)}(\theta)|\prod_{j=1}^{i-1} f^{(j)}(\theta)\prod_{l=i+1}^{m}\hat{f}^{(l)}(\theta)$$

$$\le (2D)^{m-1}\sum_{i=1}^{m}\int|f^{(i)}(\theta) - \hat{f}^{(i)}(\theta)|d\theta$$

$$\le m(2D)^{m-1}\varepsilon.$$

Letting $\hat{Z}_I = \int \prod_{i \in I} f^{(i)}$, we have

$$|Z_I - \hat{Z}_I| = \left| \int \left( \prod_{i=1}^{m} f^{(i)}(\theta) - \prod_{i=1}^{m} \hat{f}^{(i)}(\theta) \right) d\theta \right| \le \int \left| \prod_{i=1}^{m} f^{(i)}(\theta) - \prod_{i=1}^{m} \hat{f}^{(i)}(\theta) \right| d\theta \le m(2D)^{m-1}\varepsilon.$$

Thus

$$\|f_I - \hat{f}_I\|_1 = \int \left| \frac{1}{Z_I} \prod_{i=1}^{m} f^{(i)}(\theta) - \frac{1}{\hat{Z}_I} \prod_{i=1}^{m} \hat{f}^{(i)}(\theta) \right| dx = \int \left| \frac{\hat{Z}_I \prod_{i \in I} f^{(i)} - Z_I \prod_{i \in I} \hat{f}^{(i)}}{Z_I \hat{Z}_I} \right| d\theta$$

$$= \int \left| \frac{\hat{Z}_I \prod_{i \in I} f^{(i)} - \hat{Z}_I \prod_{i \in I} \hat{f}^{(i)} + \hat{Z}_I \prod_{i \in I} \hat{f}^{(i)} - Z_I \prod_{i \in I} \hat{f}^{(i)}}{Z_I \hat{Z}_I} \right| d\theta$$

$$\le \frac{1}{Z_I} \int \left| \prod_{i \in I} f^{(i)} - \prod_{i \in I} \hat{f}^{(i)} \right| d\theta + \frac{1}{Z_I} |\hat{Z}_I - Z_I|$$

$$\le \frac{2}{Z_I} m(2D)^{m-1}\varepsilon.$$

$\square$

***Proof of Theorem 4.*** Assuming $m \in [2^s, 2^{s+1})$, then after $s+1$ iterations, we will obtain our final aggregated density. At iteration $l$, each true density is some aggregation of the original $m$ densities, which can be represented by $f_{I'}$, where $I'$ is the set of indices of the original densities. Let $\varepsilon_l^{(I_1, I_2)}$ be the total variation distance between the true density and the approximation for the pair $(I_1, I_2)$ caused by combining. For example, when $l = 1$, $I_1, I_2$ contain only a single element, i.e., $I_1 = \{i_1\}$ and $I_2 = \{i_2\}$. Recall that $C_0 = \max_{I'' \subset I' \subseteq I} Z_{I''} Z_{I' \setminus I''} / Z_{I'}$, using the result from Theorem 3, we have

$$\varepsilon_1^{(I_1, I_2)} = \left\| \frac{f^{(i_1)} f^{(i_2)}}{\int f^{(i_1)} f^{(i_2)}} - \frac{\hat{f}^{(i_1)} \hat{f}^{(i_2)}}{\int \hat{f}^{(i_1)} \hat{f}^{(i_2)}} \right\|_1 \le \frac{2}{\int f^{(i_1)} f^{(i_2)}} 2D\varepsilon = \frac{2 \int f^{(i_1)} \int f^{(i_2)}}{\int f^{(i_1)} f^{(i_2)}} 2D\varepsilon \le 4C_0 D\varepsilon.$$

We prove the result by induction. Assuming we are currently at iteration $l+1$, and the paired two densities are $f_{I_1}$ and $f_{I_2}$ where $I_1, I_2 \subseteq I$. By induction, the approximation obtained at iteration $l$ are $\hat{f}_{I_1}$ and $\hat{f}_{I_2}$ which satisfies that

$$\|f_{I_1} - \hat{f}_{I_1}\|_1 \le (4C_0 D)^l \varepsilon, \quad \|f_{I_2} - \hat{f}_{I_2}\|_1 \le (4C_0 D)^l \varepsilon.$$

Using Theorem 3 again, we have that

$$\varepsilon_{l+1}^{(I_1, I_2)} = \left\| \frac{f_{I_1} f_{I_2}}{\int f_{I_1} f_{I_2}} - \frac{\hat{f}_{I_1} \hat{f}_{I_2}}{\int \hat{f}_{I_1} \hat{f}_{I_2}} \right\|_1 \le \frac{2}{\int f_{I_1} f_{I_2}} (2D) \left\{ \frac{\|f_{I_1} - \hat{f}_{I_1}\|_1 + \|f_{I_2} - \hat{f}_{I_2}\|_1}{2} \right\}$$

$$\le \frac{\int \prod_{i \in I_1} f^{(i)} \int \prod_{i \in I_2} f^{(i)}}{\int \prod_{i \in I_1 \cup I_2} f^{(i)}} (4D) \cdot (4C_0 D)^l \varepsilon \le (4C_0 D)^{l+1} \varepsilon$$

Consequently, the final approximation satisfies that

$$\|f_I - \hat{f}_I\|_1 \le (4C_0 D)^{s+1} \varepsilon \le (4C_0 D)^{\log_2 m + 1} \varepsilon.$$

$\square$

## Appendix D: Supplement to Two Toy Examples

**Bimodal Example** Figure 2 compares the aggregated density of *PART-KD/PART-ML* for several alternative combination schemes to the true density. This complements the results from one-stage combination with uniform block-wise distribution presented in Figure 1 of the main text.

Figure 2: Bimodal posterior combined from 10 subsets. The results from *PART-KD/PART-ML* multiscale histograms are shown for (1) one-stage combination with local Gaussian smoothing (2) pairwise combination with local Gaussian smoothing.

**Rare Bernoulli Example**  The left panel of Figure 3 shows additional results of posteriors aggregated from *PART-KD/PART-ML* random tree ensemble with several alternative combination strategies, which complement the results presented in Figure 2 of the main text. All of the produced posteriors correctly locate the posterior mass despite the heterogeneity of subset posteriors. The fake "ripples" produced by pairwise ML aggregation are caused by local Gaussian smoothing.

Also, the right panel of Figure 3 shows that the posteriors produced by nonparametric and semiparametric methods miss the right scale.

Figure 3: Posterior of the probability $\theta$ of a rare event combined from $M = 15$ subsets of independent Bernoulli trials. Left: the results from KD/ML multiscale histograms are shown for (1) one-stage combination with local Gaussian smoothing (2) pairwise combination with local Gaussian smoothing. Right: posterior aggregated from nonparametric and semiparametric methods.

## Appendix F: Supplement to Bayesian Logistic Regression

Figure 4 additionally plots the prediction accuracy against the length of subset chains supplied to the aggregation algorithms, for Bayesian logistic regression on two real datasets. For simplicity, the same number of posterior samples from all subset chains are aggregated, with the first 20% discarded as burn-in. As a reference, we also show the result for running the full chain. As can be seen from Figure 4, the performance of *PART-KD/ML* agrees with that of the full chain as the number of posterior samples increase, validating the theoretical results presented in Theorem 1 and Theorem 2 in the main text.

Figure 4: Prediction accuracy versus the length of subset chains on the *covertype* and the *MiniBooNE* dataset.

# References

[1] XR Chen and LC Zhao. Almost sure L1-norm convergence for data-based histogram density estimates. *Journal of Multivariate Analysis*, 21(1):179–188, 1987.