[Reviews · NeurIPS 2015]

Submitted by Assigned_Reviewer_1

[This is a "light" review.]

This paper tackles the problem of merging the results of parallel MCMC chains being run on disjoint subsets of data in large Bayesian inference problems.

It proposes an interesting alternative to existing methods, using the idea of a structured histogram.

The paper is very nicely written and gives a thoughtful and thorough treatment, looking at the various alternative approaches.

My only technical concern is the ability of the histogram-based method to scale with the dimensionality of the problem.
Summary: Proposes an interesting new idea for merging parallelized MCMC runs with data subsets.

It is a thorough and thoughtful paper.

Submitted by Assigned_Reviewer_2

This paper presents a method for aggregating samples from parallel MCMC on sub-posteriors to produce samples from the true posterior. Existing methods for aggregation, such as those presented in Consensus Monte Carlo are highly biased, whereas more sophisticated approaches such as that of Neiswanger et al and the Weierstrass sampler of Wang and Dunson, are very expensive. This paper presents a method for aggregating samples using multi-scale histograms based on random partition trees which is more scalable than existing work.

Significance: The paper addresses a very important problem: the proposed method is a way of making parallel MCMC based on subposterior aggregation more practical for Bayesian inference in large datasets.

Originality: The main novelty of the paper is a different way to aggregate samples from subposteriors into samples from the posterior. The proposed method

based on random partition trees is quite interesting, but the idea is relatively similar to using a kernel density estimate (although random partition trees seem more scalable)

Clarity: I found the paper rather hard to follow. e.g. what is j near line 109-110? Is it the j^th sample in partition A? A figure would be helpful to explain the ideas.

Quality: The method seems rather heuristic and complicated to implement, but the paper does try to analyze the error in the posterior distribution thoroughly (I did not check the proofs very well). Also they show better experimental performance to related techniques such as the Weierstrass sampler and the method of Neiswanger et al. I am not sure if the error in the method scales well with dimensionality. Finally, I wish the authors had more interesting real experiments where there is a demonstrable advantage of using Bayesian inference. e.g. for the Bayesian logistic regression case, it seems like even a point estimate computed using SGD will do well in terms of prediction accuracy.
Summary: The authors present a method for aggregating samples from parallel MCMC on sub-posteriors to produce samples from the posterior. Their method uses multiscale histograms based on random partition trees, and is more scalable than existing methods based on kernel density estimates.

Submitted by Assigned_Reviewer_3

The authors propose the use of random partition trees to aggregate the results of MCMC runs on independent subsets of data.

Given a fixed partitioning, the aggregation is done by multiplying the histogram probabilities of the partition tree and then multiplying density estimates of each chain for the samples in that partition.

This procedure is done for multiple, random partitions and the resulting density is averaged to produce the final estimated density which can finally be resampled.

Two strategies are proposed for performing the partitioning and theoretical results are provided which bound the errors made.

The method is applied to two toy problems and Bayesian logistic regression on two real datasets.

It is compared with a variety of baseline methods for sample combination.

Overall the proposed methods outperform the baselines, providing better accuracy with reduced runtime.

In general this is a strong method which has been well presented.

Summary: This paper presents a novel type of embarrassingly parallel MCMC (EPMCMC) method.

The method is simple to understan, the experiments demonstrate that it is effective and the paper is clearly written.

Submitted by Assigned_Reviewer_4

Acknowledge the author's rebuttal.

I overall maintain my general sentiment about the paper.

--

In this paper the authors describe and evaluate a method to aggregate posterior samples from an MCMC algorithm run over partitions of a dataset, where the observations are conditionally independent given the model parameters being estimated.

One of the primary limitations of existing work that the authors intend to address is that existing aggregation methods poorly handle distributions which are multimodal or have skew (i.e., those which are not Gaussian).

Their method to do this is by generating histograms of the parameter space (where the histograms are shared by partitions/samplers), then counting samples within the blocks of these histograms and aggregating across partitions by simply summing observations.

The authors then illustrate their method with several experiments - some toy problems, Bayesian logistic regression on a synthetic dataset, and some larger data sets - and illustrate that their method leads to consistently better estimates of the posterior.

What I concerns me about this paper is that I feel that the complexity of their method should be justified with more demonstration that their posterior estimate is actually better in practice than with alternative methods, especially on large datasets, for which the posterior very possibly is near-normal (more on this below).

What I like about it is that, at least as a first cut, it uses a method which is simple in theory (basically, histograms and counting) to aggregate posterior samples from disjoint partitions into a full posterior estimate.

What I think can be improved is that the complexity of the method should be better justified, and the explanation of it should be cleaned up (because while it is simple in theory, it appears complicated in practice to the extent that it is not clear that it is worth the added complexity).

More concretely, my primary suggestion for this paper to the authors is to devote more effort to describing the partition algorithms that they propose.

I believe they could do this by discussing related work a bit less (focusing mainly on the PDE equation) and by moving the remaining proofs to the appendix (but still stating the consistency results).

By doing this, the authors could make it easier for readers to implement and tune their own implementations of this algorithm.

As it is, I wouldn't want to be the grad student tasked with implementing their algorithm.

It would be worth seeing some discussion about when the assumption of skewness in a posterior with many observations (and moderate dimensionality) actually is a problem when aggregating posteriors.

Particularly, the "Bayesian central limit theorem" indicates that the posterior approaches normality when there are many samples and moderate dimensionality: http://www.stat.cmu.edu/~larry/=stat705/Lecture14.pdf.

This large-observation scenario is the exact scenario this paper is aimed to address.

When each subset has many observations, the resulting first and seconds moments may actually capture the distribution quite well, so the complexity of this method may not be justified.

In addition, I have several more comments:

- It would be good to see more discussion about previous methods for creating partition trees and how the authors' methods relate to these.

- More motivation should be given in the introduction to the motivating problems; the algorithm is jumped into without enough of a clear idea for the specific problem class.

- The "parametric" method is not described in any detail, so I am confused by what it represents.

Are these based on a parametric estimate of the distribution, where the parameter samples are aggregated with an average?

My overall takeaway from the toy examples (which exhibit skewness and/or multiple modes) is that, with averaging techniques such as mean, the resulting aggregated posterior is a poor representation of the true posterior.

However, I have questions about whether such comparisons are fair, since (at least in the case of a known bimodal distribution) averaging techniques are clearly a poor choice.

Therefore it feels that the comparison is a bit unfair.

- It might be nice to see some discussion about whether taking higher-order moments would suffice with unimodal examples. One of the columns in table 1, logistic regression, should show the likelihood of held-out observations under the posterior (e.g. averaged over posterior examples or based on the posterior mean), as optimal performance on holdout data is the most common performance metric in fitting any logistic regression.

While I don't expect the proposed method to perform worse than "average", if it doesn't then this method's complexity may not be justified for such applications.

- I'd like to see more discussion of the final examples the "Real datasets" section, as the only real discussion of it is contained in the tiny plots.

My takeaway from these is that the proposed method is able to find the best approximation to the posterior (as measured by KL divergence) while performing well - as well as the "average" method (while the average method doesn't approximate the posterior well) - on a test set.

- The authors should discuss how they selected delta_p and delta_a; was this done by fitting the model over a grid of values?

Is there a general rule of thumb for this? How poorly does their algorithm perform for poorly selected values?

Tuning these would add a significant barrier in adoption of this method.

The description of the partitioning algorithms could be cleaned up.

Further, I had no idea what sentences like "Our PART algorithm consists of space partitioning followed by density aggregation" meant on first reading them, because the terms "space partitioning" and "density aggregation" didn't yet have meeting for me (I realize that there are subsections devoted to this later in section 2).

A simple cartoon diagram would probably help significantly.

- Feel free to drop sentences like "such partitions are multi scale and related to wavelets [8]" since they are useless to people who don't already know the relation to wavelets and already known by people who do.

- The author should put much more effort into making Algorithms 1 and 2 more clear.

While I am usually pedantic about careful notation, I think pseudocode (and, as noted above, illustrations) with less emphasis on precise notation might improve the discussion here.

A highly-marked up version (with full notation) could go into the appendix.
Summary: The authors present a method for aggregating posterior samples from samplers across disjoint partitions.

They present a method which appears to work well in practice and seems simple in theory, but its complexity in practice may not be justified in many applications.

Submitted by Assigned_Reviewer_5

Comments: - What is the impact on the variable dimension on the accuracy of density estimator? - The computation cost of tree construction grows as the number of samples grow, while an accurate density estimator requires more samples. How does this impact the algorithm? - It would be interesting to include comparison with other types of MCMC methods (sub-sample or stochastic gradient based). - How can the theory on single tree be generalized to random tree ensembles, which is actually used in practice?

Updates after reading author's feedback: It would be great to include the theory on tree ensembles in the paper, which should give the reader more guidance for the real use cases(where ensemble is used)
Summary: This paper proposes to use random partition tree as density estimator for EP-MCMC. This is an interesting paper. The analysis is interesting, though it could be improved to provide theory for random tree ensembles(which was used in experiments) instead of single tree.

Author Feedback
Author rebuttal: Thanks for the careful reviews and thoughtful comments!

* Re: Reviewer 1, 2 and 6 on scalability with dimensionality & difference to KDE

Our paper is based on density estimation, where the curse of dimensionality is a long-standing issue. For a single tree, the number of blocks K grows exponentially with dimension p if every dimension is cut at least once. Given K, the number of samples required to reach below a certain error level is polynomial in K, roughly N = O(K^3) and N = O(K) respectively for ML and KD partitions, following Theorem 3 if the log terms are ignored.

We adopt histogram density estimators for efficient resampling. KDE suffers from huge resampling cost, which can kill efficiency gains in parallelization. A key observation in our paper is that if the density estimators share the same structure across subsets, resampling is greatly simplified. This technique might be extended to other estimators, which can handle higher dimensions. We offered some ways to ameliorate the dimensionality issue, including (1) averaging over random trees and (2) local Gaussian smoothing. These are useful since a poorly estimated block may overlap with other, better estimated, blocks. Sharing information across neighboring blocks (e.g. with Markov random field) could be a future extension.

* Re: Reviewer 2 and 3 on the algorithm's implementation complexity

The implementation is simpler than it seems. (We coded Algorithm 1/2 with about 200/100 lines in MATLAB). Yet, we do agree that the presentation could be significantly improved, especially for Alg 1. Alg 1 is similar to building a decision tree, except that the cutpoint is chosen by the "partition rule" \phi instead of evaluating against labels. We appreciate the suggestion that a cartoon/figure could clarify a lot. Such improvements will be included.

* Re: Reviewer 2 and 3 on real data experiments

We agree that the results in Fig. 5 only "indirectly" compare the algorithms in terms of predictive accuracy and are not the "whole story". This is because we are not able to run the full chain to convergence, due to the amount of data. In terms of prediction, we get a better accuracy-vs-time trade-off by running EP-MCMC (vs. the full chain) especially on "Covertype". We agree with Reviewer 3's suggestion that plotting with predictive (log-)likelihood is probably more informative since several curves overlap. We will replace the figures in a final version.

* Re: Reviewer 3 on the "parametric" method

The "parametric" method estimates a Gaussian to approximate each subset posterior. Then, the overall posterior is approximated by their product, with parameters computed in closed form (See line 190-191).

* Re: Reviewer 3 on asymptotic normality of the posterior

We agree that if we know the posterior is well approximated by a Gaussian, then the "parametric" method is simpler and faster. However, there are many cases where the posterior is non-Gaussian. For example, (1) In mixture models, the posterior is multi-modal due to non-identifiability of the mixture components (2) In rare event prediction, the posterior can be highly skewed and heterogeneous across subsets, even with huge n.

* Re: Reviewer 3 and 5 on toy examples

Regarding "fairness" as mentioned by Reviewer 3, we construct the toy examples as small but challenging cases for examining the accuracy of posterior aggregation. Because non-Gaussian posteriors are common in applications, a robust EP-MCMC algorithm should be able to handle all these cases. Indeed, we can expect that "average" or "parametric" will not capture the two modes. However, the failure of KDE-based methods is more thought provoking (also asked by Reviewer 5). As mentioned in our introduction, the reason is two-fold: (1) single bandwidth applied to various scales (2) poor mixing when sampling from an exponential number of mixture components (acceptance ratio is around 3% for the "bimodal" example).

* Re: Reviewer 3 on selection of parameters \delta_\rho and \delta_a

We chose them by running on a logarithmically spaced grid. If we have an idea about the scale of the posterior, \delta_a can be set to a value around or smaller than the "characteristic scale". Potentially the variables can be standardized to have a common scale before combining.

* Re: Reviewer 6 on generalizing theory to random ensembles

The theory of the random ensemble is almost identical to that of a single tree, since averaging over independent estimators does not change distributional properties. Similarly to a random forest, the improved performance mainly comes from the smoothing effect and the bias/variance trade-off. While a single tree estimator might achieve the smallest bias, it suffers from large variance, which could be reduced by averaging. A single tree estimator usually performs well on some dimensions but not all. The random ensemble improves the performance on dimensions overall.